# KNOWLEDGE-DRIVEN ACTIVE LEARNING

## ABSTRACT

The deployment of Deep Learning (DL) models is still precluded in those contexts where the amount of supervised data is limited. To answer this issue, active learning strategies aim at minimizing the amount of labelled data required to train a DL model. Most active strategies are based on uncertain sample selection, and even often restricted to samples lying close to the decision boundary. These techniques are theoretically sound, but an understanding of the selected samples based on their content is not straightforward, further driving non-experts to consider DL as a black-box. For the first time, here we propose a different approach, taking into consideration common domain-knowledge and enabling non-expert users to train a model with fewer samples. In our Knowledge-driven Active Learning (KAL) framework, rule-based knowledge is converted into logic constraints and their violation is checked as a natural guide for sample selection. We show that even simple relationships among data and output classes offer a way to spot predictions for which the model need supervision. The proposed approach (i) outperforms many active learning strategies in terms of average F1 score, particularly in those contexts where domain knowledge is rich. Furthermore, we empirically demonstrate that (ii) KAL discovers data distribution lying far from the initial training data unlike uncertainty-based strategies, (iii) it ensures domain experts that the provided knowledge is respected by the model on test data, and (iv) it can be employed even when domain-knowledge is not available by coupling it with a XAI technique. Finally, we also show that KAL is also suitable for object recognition tasks and, its computational demand is low, unlike many recent active learning strategies.

## 1 INTRODUCTION

Deep Learning (DL) methods have achieved impressive results over the past decade in fields ranging from computer vision to text generation (LeCun et al., 2015). However, most of these contributions relied on overly data-intensive models (e.g. Transformers), trained on huge amounts of data (Marcus, 2018). With the advent of Big Data, sample collection does not represent an issue any more, but, nonetheless, in some contexts the number of *supervised* data is limited, and manual labelling can be expensive (Yu et al., 2015). Therefore, a common situation is the unlabelled pool scenario (McCallumzy & Nigamy, 1998), where many data are available, but only some are annotated. Historically, two strategies have been devised to tackle this situation: semi-supervised learning which exploit the unlabelled data to enrich feature representations (Zhu & Goldberg, 2009), and active learning which select the smallest set of data to annotate to improve the most model performances (Settles, 2009).

The main assumption behind active learning strategies is that there exists a subset of samples that allows to train a model with a similar accuracy as when fed with all training data. Iteratively, the strategy indicates the optimal samples to be annotated from the unlabelled pool. This is generally done by ranking the unlabelled samples w.r.t. a given measure, usually on the model predictions (Settles, 2009; Netzer et al., 2011; Wang & Shang, 2014), or on the input data distribution (Zhdanov, 2019; Santoro et al., 2017) and by selecting the samples associated to the highest rankings (Ren et al., 2021; Zhan et al., 2021). While being theoretically sound, an understanding of the selected samples based on their content is not straightforward, in particular to non-ML experts. This issue becomes particularly relevant when considering that Deep Neural Networks are already seen as black box models (Gilpin et al., 2018; Das & Rad, 2020) On the contrary, we believe that neural models must be linked to Commonsense knowledge related to a given learning problem. Therefore, in this paper, we propose for the first time to exploit this symbolic knowledge in the selection process of an active

learning strategy. This not only lower the amount of supervised data, but it also enables domain experts to train a model leveraging their knowledge. More precisely, we propose to compare the predictions over the unsupervised data with the available knowledge and to exploit the inconsistencies as a criterion for selecting the data to be annotated. Domain knowledge, indeed, can be expressed as First-Order Logic (FOL) clauses and translated into real-valued logic constraints (among other choices) by means of T-Norms (Klement et al., 2013) to assess its satisfaction (Gnecco et al., 2015; Diligenti et al., 2017; Melacci et al., 2021).

In the experiments, we show that the proposed Knowledge-driven Active Learning (KAL) strategy (i) performs better (on average) than several standard active learning methods, particularly in those contexts where domain-knowledge is rich. We empirically demonstrate (ii) that this is mainly due to the fact that the proposed strategy allows discovering data distributions lying far from the initial training data, unlike uncertainty-based approaches. Furthermore, we show that (iii) the provided knowledge is acquired by the trained model and respected on unseen data, (iv) that KAL can also work on domains where no knowledge is available if we extract knowledge from the same model by means of a XAI technique, (v) that the KAL strategy can be easily employed also in the object-detection context, where standard uncertainty-based strategies are not-straightforward to apply (Haussmann et al., 2020) and, finally, (vi) that it is not computationally expensive, unlike many recent methods.

The paper is organized as follows: in Section 2 the proposed method is explained in details, with first an example on inferring the XOR operation and then contextualized in more realistic active learning domains; the aforementioned experimental results on different datasets are reported in Section 3, comparing the proposed technique with several active learning strategies; in Section 4 the related work about active learning and about integrating reasoning with machine learning is briefly resumed; finally, in Section 5 we conclude the paper by considering possible future work.

## 2 KNOWLEDGE-DRIVEN ACTIVE LEARNING

In this paper, we focus on classification problems with $c$ classes, in which each input $x \in X$ is associated to one or more classes and $d$ represents the input dimensionality. Let us consider the classification problem $f \colon X \to Y$, where $X \subseteq \mathbb{R}^d$ represents the feature space which may also comprehend non-structured data (e.g. input images) and $Y \subseteq \{0, 1\}^c$ is the output space composed of $c \geq 1$ dimensions. More precisely, we consider a vector function $f = [f_1, \ldots, f_c]$, where each function $f_i$ predicts the probability that $x$ belongs to the i-th class. In the Active Learning context, we also define $X_s \subset X$ as the portion of input data already associated to an annotation $y_i \in Y_s \subset Y$ and $n$ the dimensionality of the starting set of labelled data. At each iteration, a set of $p$ samples $X_p \subset X_u \subset X$ is selected by the active learning strategy to be annotated from $X_u$, the unlabelled data pool, and be added to $X_s$. This process is repeated for $q$ iterations, after which the training terminates. The maximum budget of annotations $b$ therefore amounts to $b = n + q * p$. When considering an object-detection problem, for a given class $i$ and a given image $x_j$, we consider as class membership probability, the maximum score value among all predicted bounding boxes around the objects belonging to the $i$-th class. Formally, $f_i(x_j) = \max_{s \in \mathcal{S}^i(x_j)} s(x_j)$ where $\mathcal{S}^i(x_j)$ is the set of the confidence scores of the bounding boxes predicting the $i$-th class for sample $x_j$.

Let us also consider the case in which additional *domain knowledge* is available for the problem at hand, involving relationships between data and classes. By considering the logic predicate $\mathbf{f}$ associated to each function $f$, First-Order Logic (FOL) becomes the natural way of describing these relationships. For example, $\forall x \in X$, $\mathbf{x_1}(x) \wedge \mathbf{x_2}(x) \Rightarrow \mathbf{f}(x)$, where $\mathbf{x_1}(x), \mathbf{x_2}(x)$ respectively represent the logic predicates associated to the first and the second input features, and meaning that when both predicates are true also the output function $f(x)$ needs to be true. Also, we can consider relations among classes, such as $\forall x \in X$, $\mathbf{f_v}(\mathbf{x}) \wedge \mathbf{f_z}(\mathbf{x}) \Rightarrow \mathbf{f_u}(\mathbf{x})$, meaning that the intersection between the $v$-th class and the $z$-th class is always included in the $u$-th one.

### 2.1 CONVERTING DOMAIN-KNOWLEDGE INTO LOSS FUNCTIONS

The Learning from Constraints framework (Gnecco et al., 2015; Diligenti et al., 2017) defines a way to convert domain knowledge into logic constraints and how to use them on the learning problem. Among a variety of other type of constraints (see, e.g., Table 2 in (Gnecco et al., 2015)), it studies the process of handling FOL formulas so that they can be either injected into the learning problem (in

Figure 1: A visual example of KAL working principles on the *XOR-like* problem. We depict network predictions with different colour degrees. Also, we depict in orange the samples selected by the active strategy in the current iteration and in blue those selected in previous iterations (or initially randomly annotated). Notice how the proposed method immediately discovers the data distribution not covered by the initial random sampling (right-bottom quadrant).

semi-supervised learning) or used as a knowledge verification measure (as in (Melacci et al., 2021) and in the proposed method). Going into more details, the FOL formulas representing the domain knowledge are converted into numerical constraints using the Triangular Norms (T-Norms, (Klement et al., 2013)). These binary functions generalize the conjunction operator $\wedge$ and offer a way to mathematically compute the satisfaction level of a given rule. Based on this assumption, we can detect whether the predictions made by the model on unlabelled data are coherent with the domain knowledge, or not, and possibly select these data to be annotated within an active learning strategy.

Following the previous example, $\mathbf{x_1}(x) \wedge \mathbf{x_2}(x) \Rightarrow \mathbf{f}(x)$[1] is converted into a bilateral constraint $\phi(f(x)) = 1$. By first rewriting the rule as a conjunction of terms $\neg((\mathbf{x_1} \wedge \mathbf{x_2}) \wedge \neg \mathbf{f})$[2] and by employing the product T-Norm which replace the $\wedge$ with the product operators and the $\neg \mathbf{x}$ with $1 - \mathbf{x}$, the bilateral constraint become $1 - (\mathbf{x_1}\mathbf{x_2}(1 - \mathbf{f})) = 1$. With $\varphi(f(x)) = 1 - \phi(f(x))$ we indicate the loss function associated to the bilateral constraints, which measures the level of satisfaction of the given constraints and has its minimum value in zero. Again, recalling the previous example, the associated loss function would be $\varphi(f(x)) = \mathbf{x_1}\mathbf{x_2}(1 - \mathbf{f})$, which indeed is satisfied when either $\mathbf{x_1}$ or $\mathbf{x_2}$ is zero or $\mathbf{f}$ is approximately one. For further detail on how to convert FOL formulas into numerical constraints see Appendix A.1 and Marra et al. (2019) who also proposed an an automatic computation of the loss function $\varphi$ associated to a rule. Finally, the loss function considering all the available FOL formulas $\mathcal{K}$ for the given problem is computed to drive the active strategy, selecting the points $x^\star$ which violate the most the constraints.

$$KAL: \qquad x^\star = \arg\max_{x \in X_u} \sum_{k \in \mathcal{K}} \varphi_k(f(x)) \qquad (1)$$

At each iteration, the KAL strategy selects $p$ samples $x^\star$ to annotate from the unlabelled pool $X_u$.

## 2.2 An intuitive example: the *XOR-like* problem

A well-known problem in machine learning is the inference of the eXclusive OR (XOR) operation. To show the working principles of the proposed approach, we propose a variant of this experiment, in which a neural network learns the *XOR-like* operation from a distribution of non-boolean samples. Specifically, we sampled $10^5$ points $x \in [0,1]^2$, and we assigned a label $y(x)$ as following:

$$y(x) = \begin{cases} 1, & \text{if } (x_1 > 0.5 \wedge x_2 \leq 0.5) \vee (x_1 \leq 0.5 \wedge x_2 > 0.5), \\ 0, & \text{otherwise} \end{cases} \qquad (2)$$

Also, we express the XOR operation through a FOL formula $(\mathbf{x_1} \wedge \neg \mathbf{x_2}) \vee (\neg \mathbf{x_1} \wedge \mathbf{x_2}) \Leftrightarrow \mathbf{f}$. As seen before, through the T-Norm operation we can convert the logic rule into a numerical constraint, and we can calculate its violation through the loss functions:

$$\varphi_{\mathbf{x_1} \oplus \mathbf{x_2} \rightarrow \mathbf{f}} = (\mathbf{x_1} + \mathbf{x_2} - 2\mathbf{x_1}\mathbf{x_2})(1 - \mathbf{f}),$$
$$\varphi_{\mathbf{f} \rightarrow \mathbf{x_1} \oplus \mathbf{x_1}} = \mathbf{f}(1 - (\mathbf{x_1} + \mathbf{x_2} - 2\mathbf{x_1}\mathbf{x_2})) \qquad (3)$$

---

[1]Practically, the predicate $\mathbf{x_i}(x)$ is obtained applying a steep logistic function over the input feature $x_i$: $\mathbf{x_i} = \sigma(x_i) = 1/(1 + e^{-\tau(x_i - h)})$, where $\tau$ is a temperature parameter and $h$ represents the midpoint of the logistic function itself. For predicates expressing inequalities, e.g., $\mathbf{age}(x) > 50$ we simply need to set $h = 50$.

[2]For the sake of simplicity, here and in the following, we dropped the argument $(x)$ of the logic predicates.

each one representing the loss associated to one direction of the double implication.

In Fig. 1, we reported an example of the proposed strategy starting from $n = 10$ randomly selected labelled data and by selecting $p = 5$ samples at each iteration violating the most Eq. 3, and for $q = 100$ iterations. We can appreciate how, as is often the case, the initial random sampling does not well represent the whole data distribution (no samples from the bottom-right quadrant). Nonetheless, the proposed method immediately discovers the data distribution not represented by the initial sampling (orange points—figure on the left), by selecting the samples violating $\mathbf{x_1} \oplus \mathbf{x_2} \rightarrow \mathbf{f}$. After 5 iterations (figure at the centre) the network has mostly learnt the correct data distribution. Later, the proposed strategy refines network predictions by sampling along the decision boundaries (blue points—figure on the right), allowing the network to almost already solve the learning problem (accuracy $\sim 100\%$) in just 10 iterations. As it will be seen in the next section, standard random selection (but also uncertainty-based ones) will require many more iterations.

### 2.3 REAL-LIFE SCENARIO: PARTIAL KNOWLEDGE AND DIFFERENT TYPE OF RULES

It is clear that, in the case of the *XOR-like* problem, the knowledge is complete: if we compute the predictions directly through the rule, we already solve the learning problem. However, the purpose of that simple experiment is to show the potentiality of the proposed approach in integrating the available symbolic knowledge into a learning problem. In real-life scenarios, such a situation is unrealistic, but still we might have access to some partial useful knowledge that may allow solving more quickly a given learning problem. Also, it may promote an easier interaction of domain experts with DL model by asking them to provide their knowledge rather than just data annotations.

More precisely, when we consider structured data (e.g., tabular data), a domain expert may know some very simple relations taking into consideration few features and the output class. This knowledge may not be sufficient to solve the learning problem, but a KAL strategy can still exploit it to drive the network to a fast convergence, as we will see in Section 3. On the opposite, when we consider unstructured data (e.g., images or audio signals) the employed knowledge cannot directly rely on the input features. Nonetheless, when considering a multi-label learning problem, a user may know in advance some relations between the output classes. Let us consider, as an example, a Dog-vs-Person classification: we might know that a main object (e.g., a dog) is composed of several parts (e.g., a muzzle, a body, a tail). A straightforward translation of this compositional property into a FOL rule is $\mathbf{Dog} \Rightarrow \mathbf{Muzzle} \lor \mathbf{Body} \lor \mathbf{Tail}$. Formulating the composition in the opposite way is correct as well (e.g., $\mathbf{Muzzle} \Rightarrow \mathbf{Dog}$). Also, in all classification problems, at least one of the main classes needs to be predicted (e.g., $\mathbf{Dog} \lor \mathbf{Person}$). Finally, we can always incorporate an uncertainty-like rule requiring each predicate to be either true or false, i.e., $\mathbf{Dog} \oplus \neg\mathbf{Dog}$.

## 3 EXPERIMENTS

In this work, we considered four different learning scenarios, comparing the proposed technique with several standard active strategies. We evaluated the proposed method on two standard classification problems (Bishop, 2006), the inference of the *XOR-like* problem and the classification of *IRIS* plants given their characteristics. We also considered two standard image-classification tasks: the *ANIMALS* dataset, extracted from ImageNet (Deng et al., 2009), and the Caltech-UCSD Birds-200-2011 dataset (*CUB200*, (Wah et al., 2011)). As a proof of concept, we analysed the performances of the KAL when employed in the simple *DOGvsPERSON* object recognition task. For more details regarding the latter, please refer to Appendix A.3. At last, in Appendix A.2, we report a case study showing how to apply KAL on a regression task. For each dataset, $n, p, q$, as well as the number of training epochs and the network structure are fixed in advance according to the number of classes, the dataset size and the task complexity. Reported average results are computed on the test sets of a $k$-fold Cross Validation (with $k = 10$ in the tabular data problems and $k = 5$ in the computer vision ones). More details regarding each experimental problem, as well as a table reporting all the employed rules, are available in Appendix A.4. The code will be published on a publicly available repository, and it is provided in the supplementary material. A simple code example is also reported in Appendix B showing how to solve the *XOR-like* problem with the KAL strategy. All experiments were run on an Intel i7-9750H CPU machine with an NVIDIA 2080 RTX GPU and 64 GB of RAM.

Table 1: Comparison of the methods in terms of the mean F1 score AUBC and standard deviation when increasing the number of labelled points. On top, starting and ending labelling budget for each dataset. The three best results are reported in bold. NA indicates strategies too expensive to compute.

| Dataset | XOR | IRIS | Animals | CUB200 |
|---|---|---|---|---|
| Strategy          Budget | 10-400 | 10-70 | 100-2500 | 2000-7000 |
| $\text{ADV}_{BIM}$ | $89.79_{\pm 12.14}$ | $78.81_{\pm 21.02}$ | $53.93_{\pm 0.42}$ | $50.82_{\pm 0.36}$ |
| $\text{ADV}_{DEEPFOOL}$ | $87.56_{\pm 13.08}$ | $78.86_{\pm 21.04}$ | NA | NA |
| BALD | $83.94_{\pm 9.25}$ | $90.96_{\pm 6.83}$ | $53.74_{\pm 1.96}$ | $51.17_{\pm 0.62}$ |
| KCENTER | $\mathbf{96.79}_{\pm 0.43}$ | $\mathbf{92.65}_{\pm 3.99}$ | $43.37_{\pm 3.29}$ | $48.86_{\pm 0.52}$ |
| KMEANS | $83.73_{\pm 4.40}$ | $88.55_{\pm 8.10}$ | $52.87_{\pm 1.18}$ | $49.96_{\pm 0.43}$ |
| Entropy | $90.11_{\pm 9.61}$ | $82.11_{\pm 18.49}$ | $\mathbf{57.26}_{\pm 0.94}$ | $51.91_{\pm 0.12}$ |
| $\text{Entropy}_D$ | $87.80_{\pm 11.09}$ | $81.91_{\pm 18.36}$ | $56.56_{\pm 0.95}$ | $\mathbf{51.92}_{\pm 0.39}$ |
| LeastConf | $89.97_{\pm 10.55}$ | $84.35_{\pm 13.89}$ | $53.33_{\pm 2.12}$ | $50.20_{\pm 0.43}$ |
| $\text{LeastConf}_D$ | $89.98_{\pm 12.05}$ | $84.83_{\pm 14.03}$ | $53.85_{\pm 0.84}$ | $50.07_{\pm 0.45}$ |
| Margin | $89.97_{\pm 10.55}$ | $85.46_{\pm 14.88}$ | $\mathbf{57.26}_{\pm 1.66}$ | $51.69_{\pm 0.41}$ |
| $\text{Margin}_D$ | $89.98_{\pm 12.05}$ | $84.94_{\pm 14.32}$ | $\mathbf{57.03}_{\pm 0.42}$ | $51.54_{\pm 0.48}$ |
| Random | $96.18_{\pm 0.55}$ | $92.04_{\pm 5.14}$ | $52.62_{\pm 0.49}$ | $50.29_{\pm 0.38}$ |
| SupLoss | $96.77_{\pm 0.66}$ | $92.48_{\pm 3.34}$ | $54.97_{\pm 2.10}$ | $49.53_{\pm 0.35}$ |
| KAL | $\mathbf{97.73}_{\pm 0.83}$ | $\mathbf{93.60}_{\pm 3.87}$ | $56.15_{\pm 1.04}$ | $\mathbf{51.98}_{\pm 0.35}$ |
| $\text{KAL}_D$ | $\mathbf{97.96}_{\pm 0.79}$ | $\mathbf{93.32}_{\pm 4.13}$ | $55.56_{\pm 1.21}$ | $\mathbf{52.10}_{\pm 0.24}$ |

**Compared methods** We compared KAL with 12 active learning strategies commonly considered in literature (Ren et al., 2021; Zhan et al., 2022). As representatives of uncertainty-based strategies, we considered **Entropy** (Settles, 2009) selecting samples associated to predictions having maximum entropy, **Margin** (Netzer et al., 2011) predictions with minimum margin between the top-two classes, and **LeastConf** (Wang & Shang, 2014) predictions with the lowest confidences, together with their Monte Carlo Dropout versions (Beluch et al., 2018) (respectively **Entropy**$_D$, **Margin**$_D$, **LeastConf**$_D$), which, by applying dropout a test time, compare the predictions of Monte Carlo sampled networks to better asses uncertain predictions. As more recent uncertainty-based strategies, we compared with Bayesian Active Learning by Disagreements **BALD** (Gal et al., 2017), with two strategies computing the margin by means of adversarial attacks **ADV**$_{DEEPFOOL}$ (Ducoffe & Precioso, 2018), **ADV**$_{BIM}$ (Zhan et al., 2022) and with **SupLoss** a simplified upper bound of the method proposed in (Yoo & Kweon, 2019) employing the actual labels (available only on benchmarks). As Diversity-based methods, we selected **KMeans** (Zhdanov, 2019) and **KCenter** a greedy version of the CoreSet method (Sener & Savarese, 2017). More details regarding each method are reported in Appendix A.5, together with a table resuming the associated losses.

## 3.1 KAL PROVIDES BETTER PERFORMANCE ON AVERAGE THAN ANY OTHER STRATEGY.

For a quantitative comparison of the different methods, we evaluated the performances of the different active learning strategies. In Figure 2 we reported the average F1 scores budget curves when increasing the number of selected labelled data. In Table 1 we also report the Area Under the Budget Curves (AUBC), as defined in Zhan et al. (2021). In both the *XOR-like* and *IRIS* learning problems, we can observe how both the proposed methods KAL and KAL$_D$ (the corresponding Monte-Carlo Dropout version) have the highest average AUC with a 7-12 % increment over standard uncertainty-based strategies in both cases. The only competitive method in both cases is the CoreSet-based approach KCenter. This behaviour will be better analysed in Section 3.3. Interestingly, when analysing the top plots in Fig. 2 we can appreciate how the proposed methods enable the network to learn more quickly the given tasks w.r.t. the other ones. While this was an expected behaviour on the *XOR-like* task since the provided rules completely explain the learning problem, on the *IRIS* classification task it is surprising since only 3 simple rules considering a maximum of 2 features each are given (e.g., ¬**Long_Petal** ⇒ **Setosa**). A slightly-different situation can be observed in the image classification tasks (bottom plots in Fig. 2). Here we notice the importance of employing well-structured knowledge. In the *ANIMALS* task, indeed, only 17 rules are employed relating animal species and their characteristics (e.g., **Fly** ⇒ ¬**Penguin**). In this case, the results with KAL are

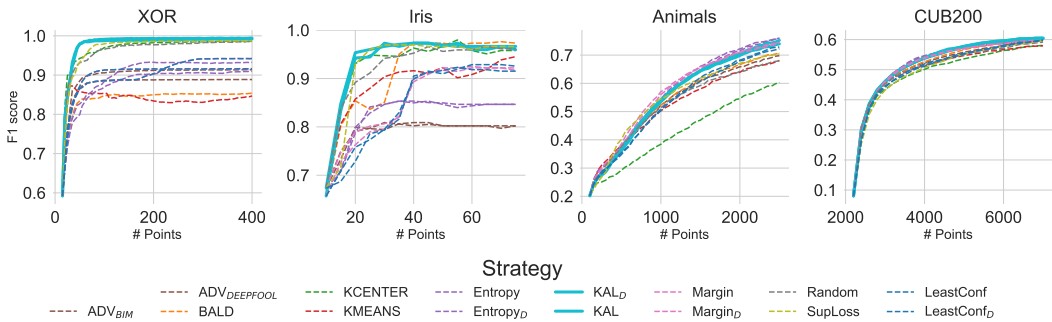

Figure 2: Average test F1 score performance growth on the four experiments when increasing the number of labelled samples. Confidence intervals have not been reported for better readability. Method variants (e.g., the Monte-Carlo Dropout versions) are displayed with the same colour.

only on average w.r.t. uncertainty-based approaches (better than LeastConf, BALD and ADV but worse than Margin and Entropy). On the contrary, KAL performs much better than KCENTER and KMEANS which are unable to correctly represents data distributions in complex scenarios. In the *CUB200* task instead, where 311 rules are employed in the KAL strategy considering bird species and their attributes (e.g., **White_Pelican ⇒ Black_Eye ∨ Solid_Belly_Pattern ∨ Solid_Wing_Pattern**), the proposed approaches are once again the best two strategies. SupLoss, instead, provide low performances in the computer vision problems. Indeed, we believe that selecting samples with high supervision loss might not be an optimal active strategy in this scenario, as it might mostly select outliers. These results prove that KAL is a very effective active learning strategy when the provided knowledge sufficiently represents the given task, both in standard and computer vision classification problems. On the *ANIMALS* task, instead, where the provided knowledge is scarce, the KAL strategy is less effective than some uncertainty strategies.

## 3.2 ABLATION STUDIES

**The amount of knowledge is directly proportional to the performance improvement** To further show the importance of having a diverse and rich set of rules as introduced in Section 3.1, we performed here an ablation study. Table 2 reports the performance of the network when equipped with a KAL strategy considering only 0%, 25%, 50%, 75% or 100% of the available knowledge. The results show evidently that the amount of knowledge is directly proportional to the performance improvement, up to $+1.8\%$. In the 0% scenario, the only rule employed is the uncertainty-like rule, which was always retained. Notice how 50.13 is similar to the LeastConf result (50.20), suggesting that KAL without any further knowledge results in an uncertainty-based strategy. In Appendix A.6, we report the complete table showing this results is valid for all experimented scenarios.

Table 2: Ablation study on the quantity of knowledge employed to support the KAL strategy on the *CUB200* dataset. The amount of knowledge is directly proportional to the increase of performance.

| Dataset | KAL 0% | KAL 25% | KAL 50% | KAL 75% | KAL 100% |
|---------|--------|---------|---------|---------|----------|
| CUB200 | $50.13_{\pm0.39}$ | $50.19_{\pm0.42}$ | $50.22_{\pm0.40}$ | $51.28_{\pm0.29}$ | $51.98_{\pm0.35}$ |

**Selecting diverse constraint violations and uncertainty-like rule improve the performances** Given a set of rules $\mathcal{K}$, the proposed method might in theory select $p$ samples all violating the same rule $\phi_k(f(x))$. To avoid this issue, we select a maximum number $r$ of samples violating a certain rule $k$, similarly to (Brinker, 2003) introducing diversity in margin-based approaches. Specifically, we group samples $x \in X_u$ according to the rule they violate the most, and we allow a maximum number of $p/2$ samples from each group (still following the ranking given by Eq. 1). In Appendix A.7, we report a table showing how requiring samples violating diverse constraints improves the overall quality of the KAL selection process. Also, we show the importance of adding the uncertainty-like rule $\bigwedge_i \mathbf{f_i} \oplus \neg\mathbf{f_i}$ introduced at the end of Section 2.3. Together, these two features allow improving the average performances of the network up to 2 %.

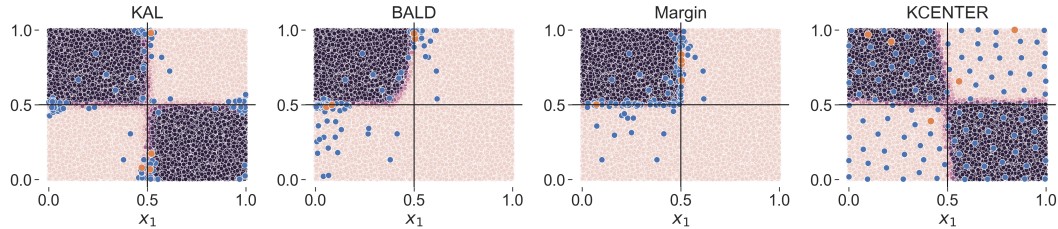

Figure 3: A comparison of the sample selection process on the *XOR-like* task at the $20^{th}$ iteration (starting from the same points as in Figure 1). Notice how both uncertainty-based strategies (BALD and Margin) have not covered yet the novel data distribution (right-bottom quadrant).

### 3.3 KAL DISCOVERS NOVEL DATA DISTRIBUTIONS UNLIKE UNCERTAINTY STRATEGIES

To further analyse the results obtained, in Figure 3 we report the samples selected by some compared strategies after 20 iteration on the *XOR-like* task (around $\sim 100$ labelled data), starting from the same randomly selected samples of Figure 1. As introduced in Section 2.2, the KAL strategy enables discovering data distribution lying far from the starting training data (leftmost figure). On the contrary, when the initial random sampling is not well representative, uncertainty-based strategies (like Margin but also BALD, central figures) are unable to discover the whole data distribution. Indeed, they require all labelled data along the decision boundaries of already known distributions. For this reason they provide mediocre results in average on *XOR-like* and *IRIS* and very high variance ($> 15 - 20\%$ on *IRIS* ). The CoreSet representative strategy, instead, has covered the four quadrants. However, by only working on input features statistics and without notion on the predictions, this strategy does not choose points along the decision boundaries, preventing the network from reaching high accuracy performances. More figures are reported in Appendix A.8.

### 3.4 KAL ENSURES DOMAIN EXPERTS THAT THEIR KNOWLEDGE IS ACQUIRED BY THE MODEL

It may be the case that domain experts are provided with a small corpus of rules which is crucial to be respected by the trained model, because, e.g., it has to be deployed in a sensitive context. By always selecting the data that violate this corpus of rules, KAL ensures them that their knowledge is aquired by the model. To simulate this scenario, we computed the argument of Eq. 1 over a small part of the CUB knowledge $\mathcal{K}_{CUB-S}$ (where $-S$ stands for small) on the test data $X_T$ for the $f^b$ model trained with all the budget: $\varphi(\mathcal{K}_{CUB-S}, f^b, X_T) = \sum_{x \in X_T} \sum_{k \in \mathcal{K}_{CUB-S}} \varphi_k(f^b(x))$. In Table 3 we report the increased percentage of the violation by models trained following a few compared methods w.r.t. the violation of a model trained following the KAL strategy and equipped with the small corpus of rules (KAL$_{small}$). The complete table together with more experimental detail is reported in appendix A.9. For the sake of completeness, this model reaches a lower test F1 AUBC (49.04). Nonetheless, it ensures domain experts that the provided knowledge is respected significantly more than using Random selection, or, worse, than standard active learning strategies.

Table 3: Violation of the $\mathcal{K}_{CUB-S}$ knowledge computed as the increased percentage over the violation of a model trained to respect this knowledge (KAL$_{small}$). The lower, the better. The proposed method ensures domain experts that their knowledge is acquired by the model.

| KAL$_{small}$ | Random | ADV$_{BIM}$ | BALD | Entropy | KCENTER | LeastConf |
|---|---|---|---|---|---|---|
| +0.00% | +483.10% | +591.03% | +720.25 | +861.74% | +530.13% | +1334.50% |

### 3.5 KAL CAN BE EMPLOYED EVEN WHEN DOMAIN-KNOWLEDGE IS NOT AVAILABLE

It might be argued that the proposed strategy can be employed only when a domain knowledge is available. However, recent works in the eXplainable AI (XAI) field (Guidotti et al., 2018; Ribeiro et al., 2018; Ciravegna et al., 2020) have shown that we can extract the same knowledge from a trained model. In general, they achieve this by training a white-box model (e.g. a decision tree) to globally explain the behaviour of a neural network. Here, we propose to employ these FOL-based

Table 4: Accuracy of the KAL strategy coupled with a XAI method, extracting the knowledge from the same network. Notice how the AUCB reduction of performance is always smaller than 1 %.

|  | XOR | IRIS | Animals | CUB |
|---|---|---|---|---|
| KAL | $97.73_{\pm 0.83}$ | $93.60_{\pm 3.87}$ | $56.15_{\pm 1.04}$ | $51.98_{\pm 0.35}$ |
| $KAL_{XAI}$ | $97.11_{\pm 0.60}$ | $92.97_{\pm 4.48}$ | $55.71_{\pm 1.24}$ | $51.38_{\pm 0.29}$ |

explanations ($\mathcal{K}_{XAI}$) as the base knowledge of the proposed strategy when no other knowledge is available ($KAL_{XAI}$). More precisely, after each iteration, we employ a simple decision tree as proposed in Guidotti et al. (2018) to extract the knowledge. More details on how we trained the XAI method are reported in Appendix A.10. However, the knowledge may be partial, particularly during the first iterations, since it is extracted on the training distribution only. Therefore, we use Eq. 1 to select only 60% of the samples, with the remaining randomly selected. This allows to eventually recover the complete knowledge. In Table 4, we report the performance of the network when equipped with this strategy ($KAL_{XAI}$), together with the performance of the standard strategy. Notice how the reduction of performance is limited to less than 1%, confirming the validity of the proposed approach even in this scenario. The amount of randomly chosen samples has not been cross-validated, therefore we expect to get even higher results by fine-tuning this parameter.

### 3.6 KAL CAN BE EMPLOYED IN OBJECT RECOGNITION TASKS

To test the proposed method in an object recognition context, as a proof of concept, we experimented on the simple *DOGvsPERSON* dataset. On this task, we compute the AUBC of the mean Average Precision curves. Also in this case, the network increases more its performances when equipped with the KAL strategy ($55.90_{\pm 0.39}$) with respect to standard random sampling ($51.41_{\pm 1.25}$) but also compared to the SupLoss method ($55.30_{\pm 0.54}$), proving the efficacy of the KAL strategy also in this context. A figure showing the three budget curves is reported in appendix A.11. Reported results are averaged over 3 initialization seeds. We only compared with Random selection and the simplified version of Yoo & Kweon (2019), since uncertainty-based strategies are not straightforward to apply in this context (Haussmann et al., 2020). Finally, we highlight again that the SupLoss performance reported is an upper bound of the performance of the method proposed in Yoo & Kweon (2019). Particularly in this context, we believe that the object recognition loss might not be easily learnt by an external model, reducing the performance of the SupLoss method.

### 3.7 KAL IS NOT COMPUTATIONALLY EXPENSIVE

When devising novel active learning techniques, of crucial importance is also the computational effort. Indeed, since re-training a deep neural network already requires a substantial amount of resources, the associated active strategy should be as light as possible. In Zhan et al. (2022), authors used as a term of comparison the average time needed to randomly sample a novel batch of data. In Table 5, we report the increased computational time in percentage. The KAL strategy (in its vanilla version) does not increase it significantly (with an average + 2-23 %). On the contrary, BALD (+ 44-191%) and, more importantly, clustering (+ 8-742%), CoreSet (2-5446%) and ADV-based (162-2451%) strategies demand considerable amounts of computational resources, strongly reducing the usability of the same methods. Standard uncertainty-based techniques like Margin, instead, are not computationally demanding, with only the Dropout versions having an increase of 2-80% (similarly to $KAL_D$). The complete table with all methods is reported in Appendix A.12.

## 4 RELATED WORK

Devising an active learning strategy is not an easy task. In the literature, two main approaches have been followed: uncertainty sampling which selects the data on which the model is the least confident; curriculum learning which instead focuses first on easy samples then extending the training set to incorporate more and more difficult ones while also targeting more diversity. Standard uncertain strategies consist in choosing samples associated to maximal prediction entropy (Houlsby et al., 2011) or at minimum distance from the hyperplane in SVM (Schohn & Cohn, 2000) or with

Table 5: Computational demand computed as the percentage increase over the time required for random sampling as defined in (Zhan et al., 2022). The lower, the better. Notice how the proposed method is less computationally expensive than many recent methods (ADV, BALD, KCENTER).

| | XOR | IRIS | Animals | CUB |
|---|---|---|---|---|
| $ADV_{BIM}$ | $+940.38\%$ $_{\pm278.11}$ | $+162.41\%$ $_{\pm31.15}$ | $+1251.36\%$ $_{\pm39.03}$ | $+1353.64\%$ $_{\pm33.53}$ |
| BALD | $+191.44\%$ $_{\pm15.14}$ | $+72.08\%$ $_{\pm47.00}$ | $+44.48\%$ $_{\pm3.34}$ | $+112.47\%$ $_{\pm13.28}$ |
| KCENTER | $+321.06\%$ $_{\pm48.54}$ | $+2.01\%$ $_{\pm1.92}$ | $+606.94\%$ $_{\pm56.57}$ | $+5446.31\%$ $_{\pm59.38}$ |
| KMEANS | $+8.30\%$ $_{\pm12.39}$ | $+9.39\%$ $_{\pm3.16}$ | $+173.20\%$ $_{\pm9.80}$ | $+742.47\%$ $_{\pm54.30}$ |
| Margin | $+0.66\%$ $_{\pm14.53}$ | $+1.85\%$ $_{\pm2.56}$ | $+1.26\%$ $_{\pm2.39}$ | $+0.65\%$ $_{\pm0.19}$ |
| $Margin_D$ | $+84.00\%$ $_{\pm13.94}$ | $+1.69\%$ $_{\pm3.90}$ | $+3.95\%$ $_{\pm3.93}$ | $+38.88\%$ $_{\pm2.49}$ |
| KAL | $+2.52\%$ $_{\pm13.69}$ | $+0.32\%$ $_{\pm2.56}$ | $+6.43\%$ $_{\pm2.02}$ | $+23.42\%$ $_{\pm0.21}$ |
| $KAL_D$ | $+88.28\%$ $_{\pm20.79}$ | $+5.41\%$ $_{\pm6.23}$ | $+8.18\%$ $_{\pm1.29}$ | $+62.98\%$ $_{\pm2.37}$ |

the highest variation ratio in Query-by-committee with ensemble methods (Ducoffe & Precioso, 2017; Beluch et al., 2018). Establishing prediction uncertainty is more difficult with DL models. Indeed, they generally tend to be over-confident, particularly when employing softmax activation functions (Thulasidasan et al., 2019). Furthermore, there is no easy access to the distance to the decision boundary as for SVM, so it needs to be computed. This problem has been tackled by devising different uncertain strategies, such as employing Bayesian Neural Network with Monte Carlo Dropout (Gal et al., 2017), by calculating the minimum distance required to create an adversarial example (Ducoffe & Precioso, 2018), or even predicting the loss associated to unlabelled sample (Yoo & Kweon, 2019). As pointed out by (Pop & Fulop, 2018), however, uncertain strategy may choose the same categories many times and create unbalanced datasets. To solve this, uncertain sample selection can be coupled with diversity sampling strategies. Diversity can be obtained by preferring batches of data maximizing the mutual information between model parameters and predictions (Kirsch et al., 2019), or selecting core-set points (Sener & Savarese, 2018), samples nearest to k-means cluster centroids (Zhdanov, 2019), or even by learning sample dissimilarities in the latent space of a VAE with an adversarial strategy (Sinha et al., 2019) or by means of a GCN (Caramalau et al., 2021).

It has been pondered that human cognition mainly consists in two different tasks: perceiving the world and reasoning over it (Solso et al., 2005). While these two tasks in humans take place at the same times, in artificial intelligence they are separately conducted by machine learning and logic programming. It has been argued that joining these tasks (to create a so-called hybrid model) may overcome some of the most important limits of deep learning, among which the *"data hungry"* issue (Marcus, 2018). In the literature, there exists a variety of proposals aiming at this objective, ranging from Statistical Relational Learning (SRL) (Koller et al., 2007) and Probabilistic Logic Programming (De Raedt & Kimmig, 2015) which focuses on integrating learning with logic reasoning, to enhanced networks focusing on relations or with external memories (Santoro et al., 2017; Graves et al., 2016). The learning from constraints framework Gnecco et al. (2015); Diligenti et al. (2017) (see survey Giunchiglia et al. (2022) for a complete list of works in this domain), instead, computes and enforce the satisfaction of a given domain knowledge within DL models. In this paper, we have shown that it can be naturally leveraged to devise active learning strategies, providing the first example of integration of symbolic knowledge in this context.

## 5 CONCLUSIONS

In this paper, we proposed an active learning strategy driven by knowledge consistency principles. The performance of a model equipped with such a strategy outperforms standard uncertainty-based approaches in context where the domain knowledge is sufficiently rich, without being computationally demanding. Furthermore, we think that KAL could induce more trust in DL with respect to standard active learning techniques by enabling non-expert users to train models leveraging their domain knowledge and ensuring them that it will be acquired by the model. A main limitation of the proposed approach is in computer vision contexts, if no attributes about the main classes are known. A possible solution could be to automatically extract such concepts from the latent space of the network, as recently proposed in Ghorbani et al. (2019); Chen et al. (2020). Also, if the domain knowledge is highly complex, FOL may not be able to fully express it and higher-order logic may be required.

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

## A    Experimental details and further results

### A.1    Converting Logic Formulas into Numerical Constraints

To better understand how the KAL framework works, in this section we will briefly go thorough some basic principles on FOL and later we will focus on how to convert rules into numerical constraints by means of different T-Norms.

**FOL Domains, Individuals, Functions, Predicates and Constraints**    In First Order Logic, a *Domain $D$* is a data space representing *Individuals $x$* which share the same representation space $X$. As an example, a domain can be composed of images representing birds, as in the CUB200 dataset. Each bird is an Individual, which is represented in a certain Domain by its features (e.g., in this case, by its image). A *Function* is a mapping of individuals between an input and an output domain. In this paper, we only focus on unary-function, i.e., functions that take only one individual in input and transform it into an individual of an output domain. An example of a function is $Age(x)$, which returns the age of a bird given its image representation. N-ary functions, taking more than one individual in input (e.g., $Relationship(x, x)$, returning the kind of relationship given two bird images) are also supported by the framework but are not used in this case. The output domain can be the same or a different domain w.r.t the input one. A *Predicate* is a special type of function returns as output a truth value $0, 1$ if we consider boolean predicate, or in $0, 1$ if also consider fuzzy values as in this work. An example of predicate is $Hummingbird(x)$, which tells you whether the considered bird belongs to the Hummingbird species. Both predicates and functions can be parametrized and learnt. In our case, all considered predicates were modelled by means of a neural network. At last, we can provide our knowledge about a certain domain by means of a set of *Constraints*. A constraint is a FOL rule defined on functions and predicates (which are the atoms of the rule). An example of a constraint could be $\forall x, Hummingbird(x) \rightarrow Bird(x)$. Finally, existential quantifier $\exists x \in X$ are also supported by the framework but we have not used them in this work.

**Converting FOL rules into numerical constraints**    To convert a FOL formula into a numerical constraints we need a way to convert connectives and quantifiers into numerical operators. To do so, we employ the fuzzy generalization of FOL that was first proposed by Novák et al. (1999). More precisely, T-norm fuzzy logic Hájek (1998) generalize Boolean Logic to continuos values in $[0, 1]$. T-norm fuzzy logics are defined by the operator modelling the AND logic operator. All the other operators are generally derived from it. In Table 6 (reported from Marra et al. (2019)), some possible implementations of common connectives when using the Product (the one used in this paper), the Lukasiewicz and the Gödel Logics.

Table 6: Some of the most used example of T-Norm with their translation of some logic operators. Table extracted from (Marra et al., 2019)

| T-norm Op. | Product | Lukasiewicz | Gödel |
|---|---|---|---|
| $x \wedge y$ | $x \cdot y$ | $\max(0, x + y - 1)$ | $\min(x, y)$ |
| $x \vee y$ | $x + y - x \cdot y$ | $\min(1, x + y)$ | $\max(x, y)$ |
| $\neg x$ | $1 - x$ | $1 - x$ | $1 - x$ |
| $x \rightarrow y$ | $1 - (x \cdot (1 - y))$ | $\min(1, 1 - x + y)$ | $x \leq y?1 : y$ |

Finally, to evaluate the violation of each numerical constraints, a loss function has to be chosen (also called *generator*). These functions need to be strictly decreasing $g : [0, 1] \rightarrow [0, +\inf]$ and such that $g(1) = 0$. Possible choices are $g(x) = 1 - x$ (as used in this paper), or $g(x) = -log(x)$.

### A.2    KAL can also be employed in Regression Task

The proposed active learning strategy can also be applied in regression tasks. In the latter, we consider a learning function $f : X \rightarrow Y$, where the output space $Y \in \mathcal{R}$ now admits values also outside the unit interval. As previously, we model $f$ with a neural network but with a single output neuron and no activation function. For what concerns the associated logic predicate, it now requires to be defined on open $e.g., \mathbf{f} > h$ or closed intervals $e.g., h_1 < \mathbf{f} < h_2$. As when defining inequalities over an

Table 7: Domain knowledge on the Insurance dataset.

| | |
|---|---|
| $\forall x$ | $\neg$ Smoker $\wedge$ Age $< 40 \Leftrightarrow$ Charge $< 7500$ |
| $\forall x$ | $\neg$ Smoker $\wedge$ Age $> 40 \Leftrightarrow$ Charge $> 7500 \wedge$ Charge $< 15000$ |
| $\forall x$ | Smoker $\wedge$ BMI $< 30 \Leftrightarrow$ Charge $> 15000 \wedge$ Charge $< 30000$ |
| $\forall x$ | Smoker $\wedge$ BMI $> 30 \Leftrightarrow$ Charge $> 30000$ |

input feature, also in this case we model the logic predicate as a continuos function by means of a sigmoid centered on $h$ – in the case of logic predicates defined over open intervals – or by means of the products of two sigmoid, the first centered on $h_1$ and the second one flipped and centered on $h_2$ – in the case of closed intervals.

To asses the validity of the proposed method on regression tasks, we experimented on the Insurance dataset available from Kaggle[3]. The proposed task is to model insurance charges based on 6 features regarding the insured persons (Age, Sex, BMI, Number of Children, Smoker and Region). The knowledge employed in this case consists of 4 rules working on 3 features and defining 4 intervals over the output space, as reported in Table 7. In Table 8 and Figure 4, instead, we reported the performance in terms of AUBC and R score when increasing the labelling budget, respectively. Also in this scenario KAL results a very effective active learning strategy. At last, uncertainty-based strategies are not reported because they cannot be applied in this scenario unless using auxiliary models to estimate confidence in this context as in (Corbiere et al., 2021).

Table 8: Comparison of the methods on a regression task in terms of R score mean AUBC and standard deviation. Budget starting ranging within 10-500 labelled points. Uncertainty based-techniques are not reported because they can not be applied in regression tasks.

| | Dataset | Insurance (R) |
|---|---|---|
| Strategy | Budget | 10-500 |
| KCENTER | | $+\mathbf{69.91}$ $_{\pm 4.96}$ |
| KMEANS | | $+58.40$ $_{\pm 8.85}$ |
| Random | | $+66.23$ $_{\pm 7.84}$ |
| SupLoss | | $+53.33$ $_{\pm 4.85}$ |
| KAL | | $+\mathbf{70.63}$ $_{\pm 6.68}$ |
| KAL$_D$ | | $+\mathbf{71.11}$ $_{\pm 6.77}$ |

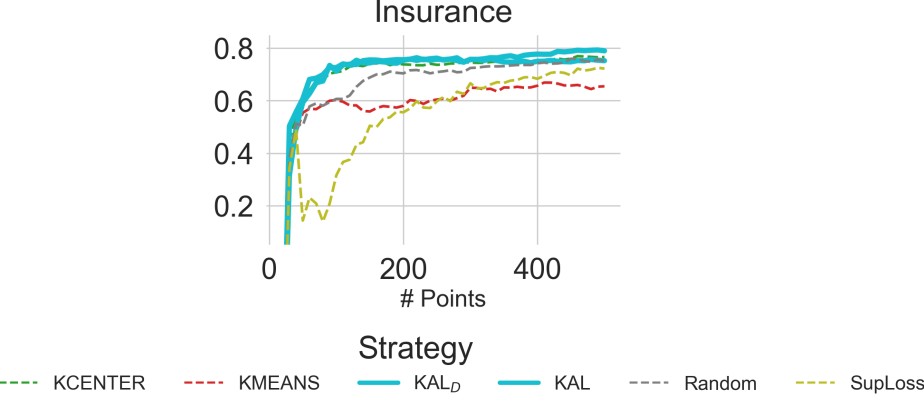

Figure 4: R Score curves when increasing the labelling budget on the Regression Insurance task.

---

[3] https://www.kaggle.com/datasets/teertha/ushealthinsurancedataset

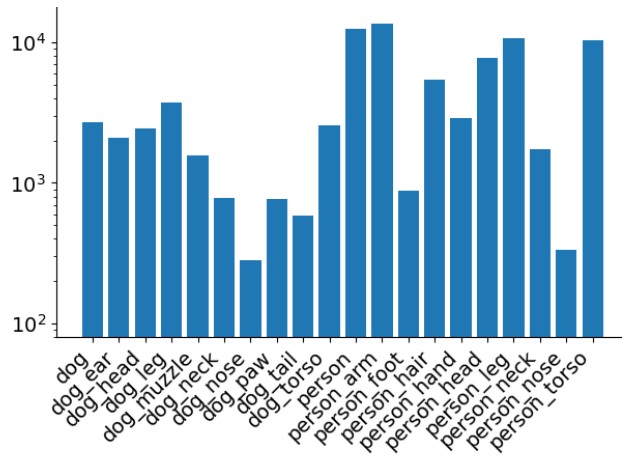

Figure 5: Class distribution in the *DOGvsPERSON* dataset.

## A.3 THE *DOGvsPERSON* DATASET

The *DOGvsPERSON* dataset is a publicly available dataset that we have created for showing the potentiality of the proposed method on a simple object-recognition problem, where well-defined relations are present among the classes, and we can employ common-knowledge rule to easily relate them. It is extracted from the *PASCAL-Part* dataset by considering only the Dog and Person main classes and their corresponding parts. In the original *PASCAL-Part* dataset, labels are given in the form of segmentation masks. We extracted a bounding box from each mask by considering the leftmost and highest pixel as the first coordinate and the rightmost and lowest pixel as the second one. Very specific parts are merged into a single class, following the approach of Serafini & d'Avila Garcez (2016) (e.g., **Left_Lower_Arm**, **Left_Upper_Arm**, **Right_Lower_Arm**, **Right_Upper_Arm** becomes **Arm**). Differently from the standard *PASCAL-Part* dataset, however, the parts in common to different objects are considered as different classes (**head** becomes **Dog_Head**, **Person_Head**). Furthermore, we only consider masks having areas $\geq 1\%$ of the whole image areas as valid label. At last, only classes appearing at least 100 times are retained. This lead to a total of 20 classes with 2 main classes (**Dog** and **Person**) and 18 parts (**Dog_Ear**, **Dog_Head**, **Dog_Leg**, **Dog_Muzzle**, **Dog_Neck**, **Dog_Nose**, **Dog_Paw**, **Dog_Tail**, **Dog_Torso**, **Person_Arm**, **Person_Foot**, **Person_Hair**, **Person_Hand**, **Person_Head**, **Person_Leg**, **Person_Neck**, **Person_Nose**, **Person_Torso**) displayed in a total of 4304 samples. Final classes are distributed in the samples as shown in Figure 5.

## A.4 NETWORK ARCHITECTURES, HYPERPARAMETERS, AND DOMAIN KNOWLEDGES

***XOR-like***    The problem of inferring the *XOR-like* operation has been already introduced in Section 2.2: it is an artificial dataset consisting of 100000 samples $x \in X \subset R^2$, mapped to the corresponding label $y \in Y \subset [0, 1]$ as in Eq. 2. A Multi-Layer Perceptron (MLP) $f \colon X \to Y$ is used to solve the task[4]. It is equipped with a single hidden layer of 100 neurons and Rectified Linear Unit (ReLU) activation, and a single output neuron with sigmoid activation. It has been trained with an AdamW optimizer Loshchilov & Hutter (2017) for 250 epochs at each iteration, with a learning rate $\eta = 10^{-3}$. Standard cross-entropy loss has been used to enforce $f$ to learn the available supervisions. By starting from $n = 10$ samples, we added $p = 5$ labelled samples at each iteration for a total of $q = 78$ iterations, resulting in a total final budget of $b = 400$ labelled samples. As anticipated, in the *XOR* problem, the rule employed for the KAL strategy is $\forall x \; \mathbf{x_1} \oplus \mathbf{x_2} \Leftrightarrow \mathbf{f}$, as also reported in Table 9.

For more real-life style problems, we have considered four more datasets where the domain-knowledge is partial or only related to the class functions.

---

[4]For all strategies requiring a multi-class output, we considered the network output as $\hat{f} = [f, 1 - f]$.

Table 9: Domain knowledge on the *XOR-like* dataset.

$$\forall x \quad \mathbf{x_1} \oplus \mathbf{x_2} \Leftrightarrow \mathbf{f}$$

**IRIS**    The *IRIS*[5] dataset is the standard iris-species classification problem. More precisely, the task consists in classifying $c = 3$ Iris species (Iris Setosa, Iris Versicolour, Iris Virginica) starting from $d = 4$ features (sepal length, sepal width, petal length, petal width). To solve the learning problem, an MLP $f \colon X^d \to Y^c$ is employed, with one hidden layer composed of 100 neurons equipped with ReLu activation functions. It has been trained again with AdamW optimizer for 200 epochs at each iteration and learning rate $\eta = 3 * 10^{-3}$. A cross-entropy loss is employed to enforce the supervisions. By starting from $n = 5$ points and by adding $p = 5$ labelled samples at each iteration for $q = 14$ iterations, leading to a total budget of $b = 75$ labelled samples. The knowledge employed in this case consists of 3 very simple rules (one per class) based on the two predicates **Long_Petal** and **Wide_Petal** (built on the $3^{rd}$ and $4^{th}$ features of the dataset, respectively, as explained in Section 2.2). In addition, in this case, a mutually exclusive rule on the classes is also considered $\forall x, \mathbf{Setosa} \oplus \mathbf{Versicolour} \oplus \mathbf{Virginica}$, as reported in Table 10.

Table 10: Domain knowledge on the *IRIS* dataset.

| | |
|---|---|
| $\forall x$ | $\neg$ Long_Petal $\Leftrightarrow$ Setosa |
| $\forall x$ | Long_Petal $\wedge \neg$ Wide_Petal $\Leftrightarrow$ Versicolour |
| $\forall x$ | Long_Petal $\wedge$ Wide_Petal $\Leftrightarrow$ Virginica |
| $\forall x$ | Setosa $\oplus$ Versicolour $\oplus$ Virginica |

**ANIMALS**    The *Animals*' dataset is a collection of 8287 images of animals, taken from the ImageNet database[6]. The task consists in the classification of 7 main classes (**Albatross**, **Giraffe**, **Cheetah**, **Ostrich**, **Penguin**, **Tiger**, **Zebra**) and 26 animal attributes (e.g., **Mammal**, **Fly** or **Lay_Eggs**), for a total of $c = 33$ classes. In this case, a Resnet50 CNN has been employed to solve the task $f \colon X^d \to Y^c$. Going into more details, a transfer learning strategy has been employed: the network $f$ has been pretrained on the ImageNet dataset (Deng et al., 2009), and two fully connected layers (the first one equipped with 100 neurons) have been trained (from scratch) on the *ANIMALS* dataset. Again, an AdamW optimizer is considered with a learning rate $\eta = 10^{-3}$ employed for 250 epochs of training at each iteration, with binary cross-entropy loss since we deal with a multi-label problem. We started with $n = 100$ labelled samples, and we added $p = 50$ samples each time for $q = 48$ iterations, for a final budget of $b = 2500$ labelled samples. In the case of *Animals*, the employed knowledge is a simple collection of 16 FOL formulas, defined by Winston & Horn (1986) as a benchmark. They involve relationships between animals and their attributes, such as $\forall x \mathbf{Fly} \wedge \mathbf{Lay\_Eggs} \Rightarrow \mathbf{Bird}$. To this collection of rules, we have also added a mutual exclusive disjunction among the animal classes (only one animal is present in each image) and a standard disjunction over the animal attributes (each animal may be associated to many attributes). The complete list of rules employed is reported in Table 11.

**CUB200**    The Caltech-UCSD Birds-200-2011[7] dataset Wah et al. (2011) is a collection of 11,788 images of birds. The task consists in the classification of 200 birds species (e.g., **Black_foooted_Albatross**) and birds attributes (e.g., **White_Throat**, **Medium_Size**). Attribute annotation, however, is quite noisy. For this reason, attributes are denoised by considering class-level annotations similarly to Koh et al. (2020). A certain attribute is set as present only if it is also present in at least 50 images of the same class. Furthermore, we only considered attributes

---

[5] *Iris*: https://archive.ics.uci.edu/ml/datasets/iris

[6] *Animals (Imagenet)*: http://www.image-net.org/, released with BSD 3-Clause "New" or "Revised" License

[7] *CUB200*: http://www.vision.caltech.edu/visipedia/CUB-200-2011 released with MIT License.

Table 11: Domain knowledge on the *Animals* dataset.

| | |
|---|---|
| $\forall x$ | Hair $\vee$ Mammal |
| $\forall x$ | Milk $\Rightarrow$ Mammal |
| $\forall x$ | Feather $\Rightarrow$ Bird |
| $\forall x$ | Fly $\wedge$ LayEggs $\Rightarrow$ Bird |
| $\forall x$ | Mammal $\wedge$ Meat $\Rightarrow$ Carnivore |
| $\forall x$ | Mamal $\wedge$ PointedTeeth $\wedge$ Claws $\wedge$ ForwardEyes $\Rightarrow$ Carnivore |
| $\forall x$ | Mammal $\wedge$ Hoofs $\Rightarrow$ Ungulate |
| $\forall x$ | Mammal $\wedge$ Cud $\Rightarrow$ Ungulate |
| $\forall x$ | Mammal $\wedge$ Cud $\Rightarrow$ Eventoed |
| $\forall x$ | Carnivore $\wedge$ Tawny $\wedge$ DarkSpots $\Rightarrow$ Cheetah |
| $\forall x$ | Carnivore $\wedge$ Tawny $\wedge$ BlackStripes $\Rightarrow$ Tiger |
| $\forall x$ | Ungulate $\wedge$ LongLegs $\wedge$ LongNeck $\wedge$ Tawny $\wedge$ DarkSpots $\Rightarrow$ Giraffe |
| $\forall x$ | Blackstripes $\wedge$ Ungulate $\wedge$ White $\Rightarrow$ Zebra |
| $\forall x$ | Bird $\wedge$ ¬Fly $\wedge$ LongLegs $\wedge$ LongNeck $\wedge$ Black $\Rightarrow$ Ostrich |
| $\forall x$ | Bird $\wedge$ ¬Fly $\wedge$ Swim $\wedge$ BlackWhite $\Rightarrow$ Penguin |
| $\forall x$ | Bird $\wedge$ GoodFlier $\Rightarrow$ Albatross |
| $\forall x$ | Albatross $\oplus$ Giraffe $\oplus$ Cheetah $\oplus$ Ostrich $\oplus$ Penguin $\oplus$ Tiger $\oplus$ Zebra) |
| $\forall x$ | Mammal $\vee$ Hair $\vee$ Milk $\vee$ Feathers $\vee$ Bird $\vee$ Fly $\vee$ Meat $\vee$ Carnivore $\vee$ PointedTeeth $\vee$ Claws $\vee$ ForwardEyes $\vee$ Hoofs $\vee$ Ungulate $\vee$ Cud $\vee$ Eventoed $\vee$ Tawny $\vee$ BlackStripes $\vee$ LongLegs $\vee$ LongNeck $\vee$ DarkSpots $\vee$ White $\vee$ Black $\vee$ Swim $\vee$ BlackWhite $\vee$ GoodFlier |

present in at least 10 classes after this refinement. In the end, 108 attributes have been retained, for a total of $c = 308$ classes. Images have been resized to a dimension $d = 256 \times 256$ pixels. The same network as in the *ANIMALS* case has been employed to solve the learning problem, with two fully connected layers trained from scratch (the first one equipped with 620 neurons – twice the dimension of the following layer). Again, an AdamW optimizer is considered with a learning rate $\eta = 10^{-3}$ for 100 epochs of training. Owing to the increased difficulty of the problem, we started with $n = 2000$ labelled samples, and we added $p = 200$ samples for $q = 25$ iterations, for a final budget of $b = 7000$ labelled samples. The knowledge employed in this case consider the relation between the classes and their attributes, with logic implications both from the class to the attributes (e.g., **White_Pelican** $\Rightarrow$ **Black_Eye** $\vee$ **Solid_Belly_Pattern** $\vee$ **Solid_Wing_Pattern**), and the vice-versa (e.g., **Striped_Breast_Pattern** $\Rightarrow$ **Parakeet_Auklet** $\vee$ **Black_throated_Sparrow** $\vee$ ...) Furthermore, a disjunction on the main classes[8] (**Black_footed_Albatross** $\vee$ **Laysan_Albatross** $\vee$ **Sooty_Albatross** $\vee$ ...) and one on the attributes are considered (**Dagger_Bill** $\vee$ **Hooked_Bill** $\vee$ **All_Purpose_Bill** $\vee$ **Cone_Bill** $\vee$ ...). A few examples of the rules employed are reported in Table 12.

***DOGvsPERSON*** This dataset has already been introduced in Appendix A.3. Since we filtered out very small object masks in this dataset, we have been able to employ a YOLOv3 model Redmon & Farhadi (2018) to solve the object-recognition problem. The model has been trained for 100 epochs at each iteration with an AdamW optimizer, with a learning rate $\eta = 3 * 10^{-4}$ decreasing by 1/3 every 33 epochs. For both training and evaluation, the Input Over Union (IOU) threshold has been set to 0.5, the confidence threshold to 0.01 and the Non-Maximum Suppression (NMS) threshold to 0.5. We started training with $n = 1000$ labelled examples and by adding $p = 500$ samples for $q = 4$ iterations for a final budget of $b = 2000$ labelled examples. In Section 3.6, we reported the AUBC of the mean Average Precision (mAP) of the model averaged 10 times with Intersection over Union (IoU) ranging from 0.5 to 0.95. On *DOGvsPERSON* we considered a set of rules listing the parts belonging to the dog or the person, (e.g., **Person** $\Rightarrow$ **Person_Arm** $\vee$ **Person_Foot** $\vee$ **Person_Hair** $\vee$ **Person_Hand** $\vee$ ...), the opposite rules implying the presence of the main object given the part (e.g., **Person_Foot** $\Rightarrow$ **Person**). Also, we considered a disjunction of all the main classes and a disjunction of all the object-parts, for a total of 22 rules employed, as reported in Table 13.

In all experiments, we employed weight decay and low learning rate to avoid overfitting, rather than employing an early stopping strategy on a separate validation set. Indeed, we argue that it is not really

---

[8]Due to the dimensionality of the dataset, the mutual exclusion of the main classes was computationally too expensive to compute in this case.

Table 12: Domain knowledge on the *CUB200* dataset. Only a few rules for each type have been reported for the sake of clarity. Also, due to the length of the rules, rules with more than two terms implied have been truncated.

| | |
|---|---|
| $\forall x$ | Black_footed_Albatross $\Rightarrow$ has_bill_shape_all-purpose $\land$ has_underparts_color_yellow $\land \dots$ |
| $\forall x$ | Laysan_Albatross $\Rightarrow$ has_bill_shape_hooked_seabird $\land$ has_breast_pattern_solid $\land \dots$ |
| $\forall x$ | Sooty_Albatross $\Rightarrow$ has_bill_shape_hooked_seabird $\land$ has_wing_color_black $\land \dots$ |
| $\forall x$ | Groove_billed_Ani $\Rightarrow$ has_bill_shape_hooked_seabird $\land$ has_breast_pattern_solid $\land \dots$ |
| $\forall x$ | Crested_Auklet $\Rightarrow$ has_wing_color_black $\land$ has_upperparts_color_black $\land \dots$ |
| | $\dots \dots$ |
| $\forall x$ | has_bill_shape_dagger $\Rightarrow$ Green_Kingfisher $\land$ Pied_Kingfisher $\land \dots$ |
| $\forall x$ | has_bill_shape_hooked_seabird $\Rightarrow$ Laysan_Albatross $\land$ Sooty_Albatross $\land \dots$ |
| $\forall x$ | has_bill_shape_all-purpose $\Rightarrow$ Black_footed_Albatross $\land$ Red_winged_Blackbird $\land \dots$ |
| $\forall x$ | has_bill_shape_cone $\Rightarrow$ Parakeet_Auklet $\land$ Indigo_Bunting $\land \dots$ |
| $\forall x$ | has_wing_color_brown $\Rightarrow$ Brandt_Cormorant $\land$ American_Crow $\land \dots$ |
| | $\dots \dots$ |
| $\forall x$ | Black_footed_Albatross $\lor$ Laysan_Albatross $\lor$ Sooty_Albatross $\lor \dots$ |
| $\forall x$ | has_bill_shape_dagger $\lor$ has_bill_shape_hooked_seabird $\lor$ has_bill_shape_all-purpose $\lor \dots$ |

Table 13: Domain knowledge on the *DOGvsPERSON* dataset.

| | |
|---|---|
| $\forall x$ | Dog_ear $\Rightarrow$ Dog |
| $\forall x$ | Dog_head $\Rightarrow$ Dog |
| $\forall x$ | Dog_leg $\Rightarrow$ Dog |
| $\forall x$ | Dog_muzzle $\Rightarrow$ Dog |
| $\forall x$ | Dog_neck $\Rightarrow$ Dog |
| $\forall x$ | Dog_nose $\Rightarrow$ Dog |
| $\forall x$ | Dog_paw $\Rightarrow$ Dog |
| $\forall x$ | Dog_tail $\Rightarrow$ Dog |
| $\forall x$ | Dog_torso $\Rightarrow$ Dog |
| $\forall x$ | Person_arm $\Rightarrow$ Person |
| $\forall x$ | Person_foot $\Rightarrow$ Person |
| $\forall x$ | Person_hair $\Rightarrow$ Person |
| $\forall x$ | Person_hand $\Rightarrow$ Person |
| $\forall x$ | Person_head $\Rightarrow$ Person |
| $\forall x$ | Person_leg $\Rightarrow$ Person |
| $\forall x$ | Person_neck $\Rightarrow$ Person |
| $\forall x$ | Person_nose $\Rightarrow$ Person |
| $\forall x$ | Person_torso $\Rightarrow$ Person |
| $\forall x$ | Dog $\Rightarrow$ Dog_ear $\lor$ Dog_head $\lor$ Dog_leg $\lor$ Dog_muzzle $\lor$ Dog_neck $\lor$ Dog_nose $\lor$ Dog_paw $\lor$ Dog_tail $\lor$ Dog_torso |
| $\forall x$ | Person $\Rightarrow$ Person_arm $\lor$ Person_foot $\lor$ Person_hair $\lor$ Person_hand $\lor$ Person_head $\lor$ Person_leg $\lor$ Person_neck $\lor$ Person_nose $\lor$ Person_torso |
| $\forall x$ | Dog $\lor$ Person |
| $\forall x$ | Dog_ear $\lor$ Dog_head $\lor$ Dog_leg $\lor$ Dog_muzzle $\lor$ Dog_neck $\lor$ Dog_nose $\lor$ Dog_paw $\lor$ Dog_tail $\lor$ Dog_torso $\lor$ Person_arm $\lor$ Person_foot $\lor$ Person_hair $\lor$ Person_hand $\lor$ Person_head $\lor$ Person_leg $\lor$ Person_neck $\lor$ Person_nose $\lor$ Person_torso |

realistic to rely on sufficiently large validation sets in an active learning scenario, where the amount of labels is scarce, and we try to minimize it as much as possible. Also, rather than retraining the network from scratch at each iteration, we choose to keep training it to save computational time.

## A.5 COMPARED METHOD DETAILS

In the experiments, several methods have been evaluated, comparing the performances both in terms of F1 score improvement and in terms of selection time. Compared techniques mostly follow two different active learning philosophies: uncertainty sample selection, which aim at estimating

Table 14: Resume of the active losses $\mathcal{L}_a(f, x)$ maximized by each of the compared methods. Variants of the same methods (e.g., employing Monte Carlo Dropouts) have not been reported for the sake of brevity. We indicate with $H$ the Entropy, with $\sigma$ the softmax activation, with $p_f(y_i, x)$ the probability of associated to the class $i$, with $M$ the set of main classes, with $\epsilon$ the adversarial perturbation, $g(\cdot)$ the model used to learn the model loss, with $Z_{CENTER}$ the CoreSet points and with $Z_{MEANS}$ the centroids of K-Means.

| | |
|---|---|
| Entropy | $\mathcal{L}_a = H[f(x)|x] = -(\sigma(f_M(x)) \cdot \log(\sigma(f_M(x))))$ |
| Margin | $\mathcal{L}_a = -(p_f(\hat{y}_1|x) - p_f(\hat{y}_2|x)), \quad \hat{y}_1, \hat{y}_2 = \arg\max^2_{i \in M} p_f(y_i, x)$ |
| LeastConf | $\mathcal{L}_a = 1 - p_f(\hat{y}|x), \quad \hat{y} = \arg\max_{i \in M} p_f(y_i, x)$ |
| BALD | $\mathcal{L}_a = H[f(x)|x] - \mathbb{E}[H[f(x)|x, \omega]$ |
| ADV | $\mathcal{L}_a = ||\epsilon||, \quad \epsilon : f_M(x + \epsilon) \neq f_M(x)$ |
| SupLoss | $\mathcal{L}_a = \mathcal{L}(f(x), y) \sim g(f(x))$ |
| KCENTER | $\mathcal{L}_a = ||x - z_k||, \quad z_k = \arg\min_{z \in Z_{CENTER}} ||x - z||$ |
| KMEANS | $\mathcal{L}_a = -||x - z_k||, \quad z_k = \arg\min_{z \in Z_{MEANS}} ||x - z||$ |
| KAL | $\mathcal{L}_a = \sum_{k \in \mathcal{K}} \varphi_k(f(x))$ |

prediction uncertainty in deep neural networks; and diversity selection, which aims at maximally covering the input data distribution. Several standard strategies like **Entropy** (Settles, 2009), **Margin** (Netzer et al., 2011) and **LeastConf** (Wang & Shang, 2014) belong to the first group, as well as some more recent methods like **BALD** (Gal et al., 2017), **ADV**$_{DEEPFOOL}$ (Ducoffe & Precioso, 2018), **ADV**$_{BIM}$ (Zhan et al., 2022) and **SupLoss** (Yoo & Kweon, 2019). On the contrary, **KMeans** (Zhdanov, 2019) and KCenter (Sener & Savarese, 2017), aim at reaching the highest diversity among selected samples.

In Table 14, we reported the loss $\mathcal{L}_a(f, x)$ used by each method to select active samples as following:

$$x^\star = \arg\max_{x \in X_U} \mathcal{L}_a(f, x) \tag{4}$$

As the name implies, the **Entropy** method aims at measuring the uncertainty of a prediction by means of its entropy. With $\sigma$ we indicate the softmax activation, that we applied on top[9] of $f_M$ to obtain a probability distribution – i.e., $\sum_{i \in M} \sigma(f_i, x) = 1$. – as required to compute the Entropy. To adapt this method to the multi-label context, we restricted the computation to the $M$ main classes (mutually exclusive) of $f$. Similarly, in **Margin** and **LeastConf** we compute again the softmax over the main classes $f_M$ to obtain for each main class $i$ the probability $p_f(y_i, x)$ of being active. In the first case, the margin between the two classes associated to the highest probability is used to estimate the uncertainty of the prediction. In the second case, the inverse of the probability associated to the most probable class is used instead. **BALD** employs as acquisition function the mutual information between the model predictions and the model parameters. The first term of the loss is again the Entropy of the model predictions; the second term of the equation, instead, represents the expected value of the entropy over the posteriori of the model parameters. Basically, the overall value is high when the predictions differ a lot, but the confidence of each prediction is rather high. Both **ADV**$_{DEEPFOOL}$ and **ADV**$_{BIM}$ use as a metric of uncertainty the norm of $\epsilon$, the minimum input alteration allowing to change the prediction of the network. The difference in the two methods relies on the way $\epsilon$ is computed (i.e., on the adversarial attack employed). The stopping criterion to find $\epsilon$ in both methods consists in finding a perturbation that induce the network to predict a different class. To adapt these methods to the multi-label context, therefore, we restricted again the classes to the main ones $f_M$. The **SupLoss** strategy, instead, aims at approximating the supervision loss by means of a model $g(f(x))$. It receives in input the output of the $f$ network (as well as the activation of the last hidden layers) and is trained to mimic the actual loss $\mathcal{L}(f(x), y) \sim g(f(x))$ on the supervised data. As Yoo & Kweon (2019) claim that $\mathcal{L}(f(x), y)$ is the actual upper bound of their method, for simplicity we employed the same $\mathcal{L}$ to select uncertain samples. However, as we have seen in Section 3.1, even in the best case scenario this strategy does not work very well in complex problem as it mostly end up selecting outliers and making network convergence more difficult. Both **KCENTER** and **KMEANS**, instead, base their selection criteria only on distance metric on the input

---

[9]To avoid numerical issues, we applied it on the logits of the output of the network $log(f(x)) - log(1 - f(x))$.

| Strategy | XOR | IRIS | Animals | CUB200 |
|----------|-----|------|---------|--------|
| KAL 00 % | $93.54_{\pm 0.83}$ | $81.45_{\pm 18.88}$ | $53.42_{\pm 1.64}$ | $50.13_{\pm 0.36}$ |
| KAL 25 % | – | $91.72_{\pm 5.03}$ | $53.21_{\pm 1.39}$ | $50.22_{\pm 0.40}$ |
| KAL 50 % | – | $93.12_{\pm 4.72}$ | $54.54_{\pm 1.11}$ | $50.19_{\pm 0.42}$ |
| KAL 75 % | – | $\mathbf{93.84}_{\pm 3.94}$ | $54.61_{\pm 0.47}$ | $51.28_{\pm 0.29}$ |
| KAL 100 % | $\mathbf{97.73}_{\pm 0.83}$ | $93.60_{\pm 3.87}$ | $\mathbf{56.15}_{\pm 1.06}$ | $\mathbf{51.98}_{\pm 0.31}$ |

Table 15: Complete version of Table 2. The amount of knowledge is proportional to performance improvement. In the XOR dataset, since we only used the XOR rule, we only reported

data distribution [10]. **KCENTER** aims at covering as much as possible the input data distribution, by selecting at each iteration the furthest sample to the current set of labelled samples ($Z_{CENTER}$). On the contrary, **KMEANS** strategy selects the closest unlabelled sample to the set of centroids ($Z_{MEANS}$), following a curriculum-learning strategy.

## A.6 ABLATION STUDY: THE AMOUNT OF KNOWLEDGE IS PROPORTIONAL TO PERFORMANCE IMPROVEMENT

We here report the complete version of Table 2. The result that we reported in the main paper for the *CUB200* dataset is valid for all datasets: the amount of knowledge is proportional to the performance improvement. In the XOR dataset, since we only had the XOR rule, we only reported the performance with 0 % and 100 % knowledge. We recall that in the 0 % knowledge scenario (to still evaluate our method) we always retain the uncertainty-like rule.

## A.7 ABLATION STUDY: SELECTING DIVERSE CONSTRAINT VIOLATIONS AND EMPLOYING AN UNCERTAINTY-LIKE RULE IMPROVES THE PERFORMANCES

In this section, we analyse the role of selecting samples violating a diverse set of rules and of employing the uncertainty-like rule introduced in Sec. 3.2. We recall that diversity sample selection is achieved by requiring a maximum of $r = p/2$ samples violating a certain rule $k$ when selecting a new batch of $p$ samples to be labelled. The uncertainty-like rule, instead, consists in requiring each predicate $\mathbf{f_i}$ to be either true or false $\bigwedge_i \mathbf{f_i} \oplus \neg \mathbf{f_i}$. This way, in case many predicates for a certain sample have a not well-defined value (e.g., $f_i(x) = 0.5$), the violation of the constraint associated to this rule will be high.

We studied three different scenarios, employing the uncertainty-like rule (Unc), requiring a set of sample violating different rules (Div) or both of them (Div Unc). We compared them with the plain versions both in the case of estimating predictions with a standard classifier (KAL) and when using Monte Carlo dropout ($KAL_D$), leading to a total set of eight configurations. We report the results of these ablation study in Table 16. The results reported as KAL in Section 3.1 are, actually, the version with both features (KAL Div Unc). Indeed, KAL Div Unc and $KAL_D$ Div Unc respectively increase the AUBC by 2.3% and 1.9% on the Animals dataset over KAL and $KAL_D$; on the *XOR-like* and on the *CUB200* dataset, the increase is as well important, although smaller than in the latter case; only in the *IRIS* case, the performances get reduced when requiring samples violating a diverse set of rules. This can be most likely explained by considering the size of the Iris dataset and the fact that we employed only 3 rules, specific for each class. In this setting, it may be more convenient for the network to gather samples related to a specific class (i.e., violating a specific rule) only. Indeed, labelling points related to a distribution already covered (i.e., where the knowledge is mostly respected) with very few examples available can slightly decrease the overall performances.

## A.8 TRAINING EVOLUTIONS ON THE *XOR-like* PROBLEM

In Figure 6 we report further snapshots of the active selection process on the *XOR-like* dataset. They depict the model predictions similarly to Figure 3, but at different iterations and for all the

---

[10]In the image recognition tasks, we employed the latent distribution extracted by the convolutional features, as commonly done (Ren et al., 2021; Zhan et al., 2022).

| Strategy | XOR | Iris | Animals | CUB200 |
|---|---|---|---|---|
| KAL | $97.38_{\pm0.40}$ | $\mathbf{94.01}_{\pm4.06}$ | $53.86_{\pm1.79}$ | $51.94_{\pm0.43}$ |
| KAL Unc | $97.46_{\pm0.40}$ | $93.97_{\pm3.61}$ | $54.12_{\pm0.97}$ | $51.98_{\pm0.35}$ |
| KAL Div | $97.62_{\pm0.79}$ | $93.44_{\pm4.50}$ | $54.56_{\pm1.09}$ | $51.97_{\pm0.39}$ |
| KAL Div Unc | $97.73_{\pm0.83}$ | $93.60_{\pm3.87}$ | $\mathbf{56.15}_{\pm1.04}$ | $51.98_{\pm0.35}$ |
| $KAL_D$ | $97.45_{\pm0.49}$ | $\mathbf{94.11}_{\pm4.23}$ | $53.67_{\pm1.64}$ | $51.93_{\pm0.47}$ |
| $KAL_D$ Div | $\mathbf{97.96}_{\pm0.79}$ | $93.32_{\pm4.14}$ | $53.74_{\pm1.42}$ | $51.90_{\pm0.44}$ |
| $KAL_D$ Unc | $97.57_{\pm0.51}$ | $93.51_{\pm3.80}$ | $53.92_{\pm1.28}$ | $\mathbf{52.02}_{\pm0.33}$ |
| $KAL_D$ Div Unc | $\mathbf{97.96}_{\pm0.79}$ | $93.32_{\pm4.13}$ | $\mathbf{55.56}_{\pm1.21}$ | $\mathbf{52.10}_{\pm0.24}$ |

Table 16: Ablation study on diverse rule selection (**Div**) and on the uncertainty-like rule (**Unc**), both on the standard KAL and on Monte Carlo dropout version ($KAL_D$). In bold, the two best results. Notice how, the latter increase the AUBC by 2.3% and 1.9% respectively over KAL and $KAL_D$, on the Animals dataset.

compared methods. In this figure, it is even more clear that *no* uncertainty-based strategy is capable of discovering novel data distributions. More precisely, nor Entropy, nor Margin, nor LeastConf, nor BALD, nor $Adv_{BIM}$, nor $Adv_{DEEPFOOL}$ can cover the data distribution in the right-bottom angle (for which no samples have been drawn during the initial random sampling) even after 20 iterations. Diversity-based strategy, instead, can cover all data distribution; however, both KCENTER and even more KMEANS cannot perfectly predict samples along the decision boundaries. Finally, it is interesting to notice how the sampling selection performed by the SupLoss resemble the one made by KAL. However, even in the simplified upper case considered in this comparison, the selection process performed by SupLoss drives the network more slowly to convergences with respect to KAL. Indeed, as it can be noticed in the right-most plots, after $20^{th}$ iterations the selection process in SupLoss has not covered very well yet the centre, differently from the selection process of KAL, which correctly covered this important zone.

### A.9  COMPARISON OF THE KNOWLEDGE VIOLATION

As introduced in Section 3.4, we tested whether the proposed method allows the model to learn to respect the knowledge provided by domain experts. In Table 17 we reported an extended version of Table 3, computing the increased percentage of violation of the $\mathcal{K}_{CUB-S}$ knowledge by models trained following the compared active strategies w.r.t. a model trained following the proposed strategy ($KAL_{SMALL}$). All the compared methods (including KALs when equipped with all the knowledge on CUB) violate the $\mathcal{K}_{CUB-S}$ knowledge significantly more (4-13 times) than ($KAL_{SMALL}$).

### A.10  EXTRACTING THE KNOWLEDGE WITH AN XAI METHOD

Explainable AI techniques are more and more used in literature to mitigate the intrinsic opacity of deep neural networks. In Sec. 3.5 we employed Guidotti et al. (2018) to explain the $f$ model at each iteration, and use the explanations as the base knowledge ($\mathcal{K}_{XAI}$) of the proposed method ($KAL_{XAI}$). This is a viable solution to employ when no other knowledge is available. More precisely, we trained a decision tree $h$ on the supervised training input data $X_s$ to mimic the behaviour of the network $f$, i.e., we minimized the following loss $\mathcal{L}_{X_s}(h(x), f(x))$. From the decision tree, we extracted global explanations of each class in the form of, e.g., $\forall x$ Setosa $\Leftrightarrow \neg$ Long_Petal. In computer vision tasks, however, decision trees are not suitable to be employed directly on the raw input data. For this reason, following Barbiero et al. (2022), we trained the decision tree to mimic the behaviour of the model $f$ over the main classes when receiving in inputs the attribute ones. The extracted rules, in this case, are of the type Black_footed_Albatross $\Rightarrow$ has_bill_shape_all-purpose $\wedge$ has_underparts_color_yellow $\wedge$ …. To also explain the attribute classes, we did the reverse, i.e. we trained the decision tree to mimic the behaviour of the model $f$ over the attribute classes and receiving in input the main ones. The extracted rules in this case are of the type has_bill_shape_cone $\Rightarrow$ Parakeet_Auklet $\wedge$ Indigo_Bunting $\wedge$ …. Finally, we always added to this set of rule a mutual exclusion rule over the main classes and a disjunction over the attributes (where available). Indeed,

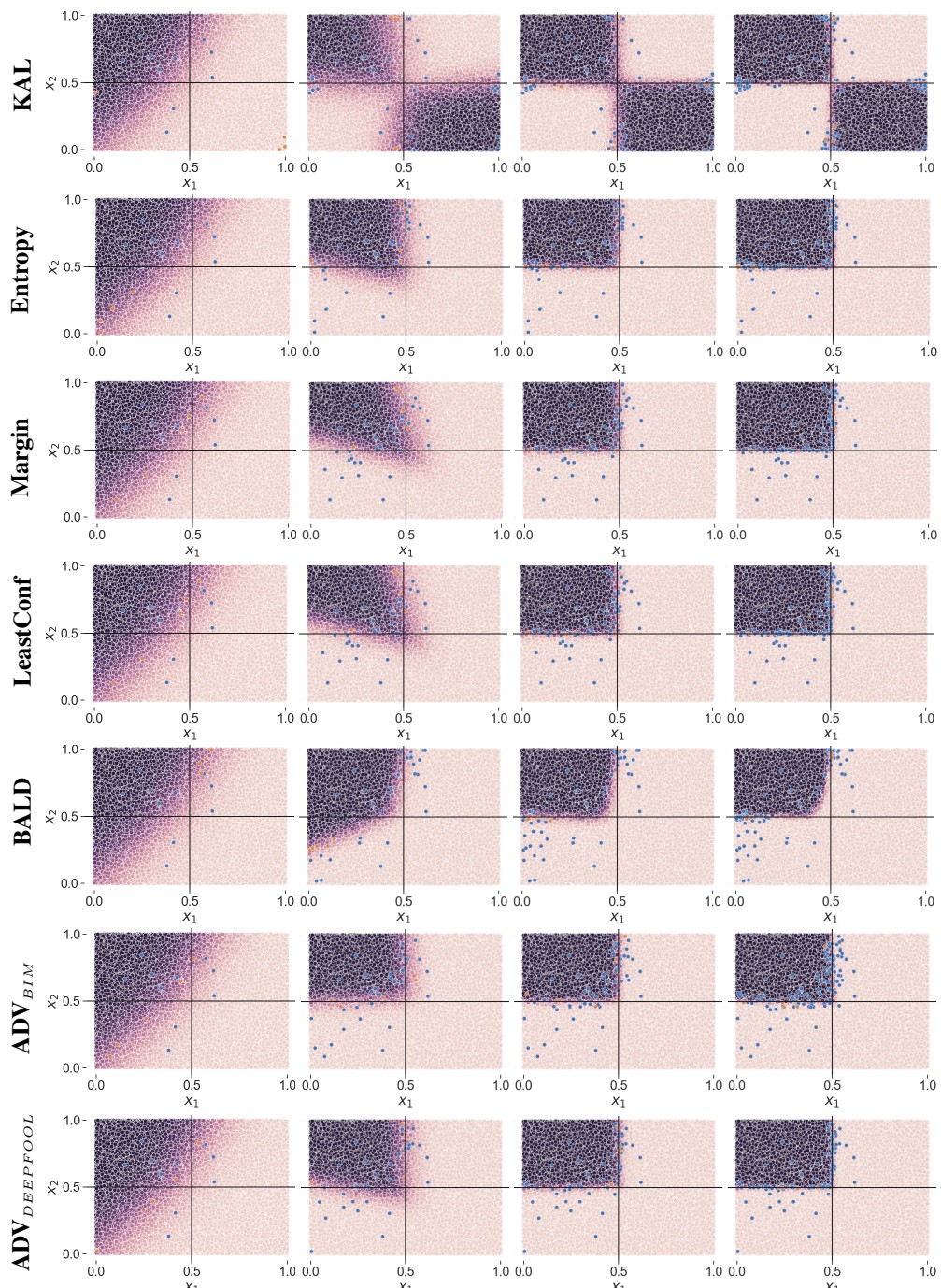

Figure 6: An illustration of the selection process evolution for all the compared strategies on the *XOR-like* problem. We depict network predictions with different colour degrees (light colours negative predictions, dark colours positive prediction); in blue, samples selected in previous iterations, in orange those selected at the current iteration. Black lines at $x_1 = 0.5$ and $x_2 = 0.5$ are reported for visualization purposes only. From left to right, the state at the $1^{st}$, $5^{th}$, $10^{th}$ and $20^{th}$ iteration.

this notion must be always available as it conditions the choice of the training loss (e.g. binary vs standard cross entropy).

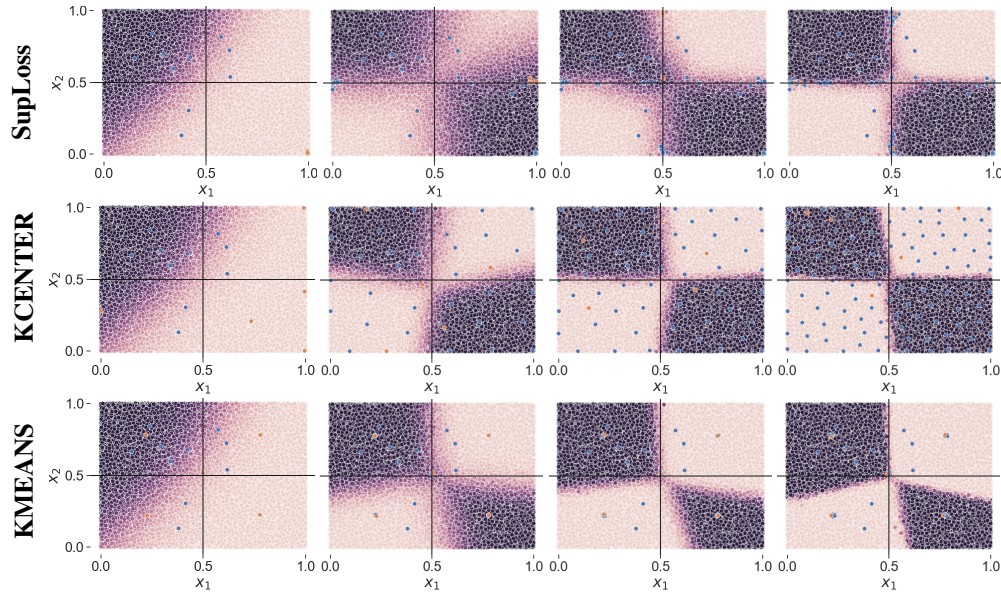

Figure 6: A visual example on the *XOR-like* problem, showing how the training evolves in each of the compared strategy (continued from previous page).

Table 17: Violation of the $\mathcal{K}_{CUB-S}$ knowledge computed as the increased percentage over the violation of a model trained to respect this knowledge (KAL$_{small}$). The proposed method ensures domain experts that their knowledge is acquired by the model on test data significantly more than using standard techniques.

| Strategy | Increased Violation |
|---|---|
| KAL$_{SMALL}$ | 0.00 $\pm_{38.09}$ % |
| ADV$_{BIM}$ | 591.03 $\pm_{532.47}$ % |
| BALD | 720.25 $\pm_{251.86}$ % |
| Entropy | 861.74 $\pm_{519.35}$ % |
| Entropy$_D$ | 812.35 $\pm_{359.23}$ % |
| KAL | 863.53 $\pm_{226.81}$ % |
| KAL$_D$ | 890.98 $\pm_{299.79}$ % |
| KCENTER | 530.13 $\pm_{188.74}$ % |
| KMEANS | 555.23 $\pm_{327.80}$ % |
| LeastConf | 1334.50 $\pm_{697.47}$ % |
| LeastConf$_D$ | 934.46 $\pm_{505.21}$ % |
| Margin | 846.05 $\pm_{341.52}$ % |
| Margin$_D$ | 637.26 $\pm_{165.97}$ % |
| Random | 483.10 $\pm_{285.08}$ % |
| SupLoss | 804.38 $\pm_{76.31}$ % |

### A.11 EMPLOYING ACTIVE STRATEGIES IN OBJECT RECOGNITION TASK

As introduced in Section 3.6, we tested the proposed method also in an object recognition context. As a proof of concept, we experimented on the simple DOGvsPERSON dataset, described in Appendix A.3. In Table 18, we report again the AUBC of the mean Average Precision curves when increasing the budget of labelled points. The same curves are reported in Figure 7. The results in this case are averaged over three different seed initialization of the network. The network increases its more performances when equipped with the KAL strategy (55.90) with respect to standard random sampling (51.41) but also compared to the SupLoss method (55.30). Interestingly, by considering Figure 7, we can appreciate how also in this context the proposed strategy significantly improves the performance of the network already after the very first iterations. Finally, we highlight again that the SupLoss performance reported are an upper bound of the performance of the method proposed in (Yoo

| Dataset | # Points | Random | SupLoss | KAL |
|---|---|---|---|---|
| Dog vs Person | 1000-3000 | 51.27 $_{\pm1.41}$ | 55.30 $_{\pm0.54}$ | **55.90** $_{\pm0.39}$ |

Table 18: Comparison of the methods in terms of the test mAP (%) AUBC (Zhan et al., 2021) when increasing the number of labelled points on the object recognition task.

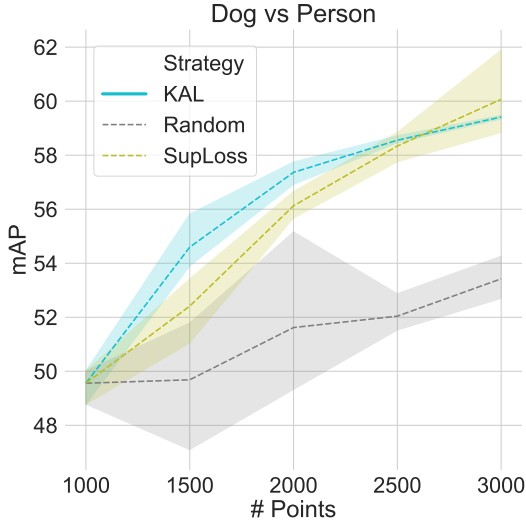

Figure 7: Average test mAP performance growth on the object recognition task when increasing the number of labelled samples. Shadowed areas indicate the 95 % confidence interval.

& Kweon, 2019). Particularly in this context, we believe that the object recognition supervision loss might not be easily learnt by the external model $g$. In this proof-of-concept, we only compared with Random and our implementation of the SupLoss method, since they were the most straightforwardly adaptable method to the object recognition context. In future work, we plan to compare with the adapted uncertainty-based strategy proposed in Haussmann et al. (2020).

## A.12   Experimental time comparison

In Table 19, we report the complete comparison of the time required by the different strategies. More precisely, we reported the increased percentage in computational time. We included both the time required to select the data and the time required to train the network. The KAL strategy does not come at the cost of a significantly increased time (with an average +0.3-23.4 %), comparably with standard uncertainty based strategies, Entropy (+0.0-32.9%), LeastConf (+0.0-2.01%), and Margin (+0.6-1.8%). Only the Monte Carlo Dropout versions, require higher computational time (KAL$_D$ +5.4-88.3%, Entropy$_D$ +0.0-89.2%, LeastConf$_D$ +0.3-36.6%, Margin$_D$ +1.7-84.0%) since they repeat several times a prediction with different switched-off neurons, to better assess uncertainty. On the contrary, BALD (+44.5-191.4%) and, more importantly, KMeans (+8.3-742.5%), KCenter (+2.0-5446.3%), ADV$_{BIM}$ (+162.4-1353.6%), and ADV$_{DEEPFOOL}$ (+206.8-NA) strategies demand remarkable computational resources, strongly reducing the usability of the same methods. In particular, ADV$_{DEEPFOOL}$ requires a huge amount of resource, since the DEEPFOOL attack is linearly dependent in both the selected samples and the predicted classes. For this reason, it was computationally infeasible to experimented it on the Animals and CUB datasets (respectively with 7 and 200 classes). At last, we reported SupLoss with a $^\star$, since the computational time required to compute the cross-entropy loss between the labels and the prediction (as simplified in this comparison), may be significantly different from the one required to predict the loss through the model $g$.

Table 19: Computational time required to select the samples to annotate by all the compared methods as percentage increase w.r.t the time required for a random sampling as defined in (Zhan et al., 2022). The lower, the better. NA indicates strategies computationally too expensive to compute on certain datasets. Notice how the proposed method is not computationally expensive, contrarily many recent active learning methods proposed in the literature.

| Strategy | XOR | IRIS | Animals | CUB |
|---|---|---|---|---|
| $\text{ADV}_{BIM}$ | $+940.38\%\ _{\pm 278.11}$ | $+162.41\%\ _{\pm 31.15}$ | $+1251.36\%\ _{\pm 39.03}$ | $+1353.64\%\ _{\pm 33.53}$ |
| $\text{ADV}_{DEEPFOOL}$ | $+2450.66\%\ _{\pm 1435.04}$ | $+206.79\%\ _{\pm 110.24}$ | NA | NA |
| BALD | $+191.44\%\ _{\pm 15.14}$ | $+72.08\%\ _{\pm 47.00}$ | $+44.48\%\ _{\pm 3.34}$ | $+112.47\%\ _{\pm 13.28}$ |
| KCENTER | $+321.06\%\ _{\pm 48.54}$ | $+2.01\%\ _{\pm 1.92}$ | $+606.94\%\ _{\pm 56.57}$ | $+5446.31\%\ _{\pm 59.38}$ |
| KMEANS | $+8.30\%\ _{\pm 12.39}$ | $+9.39\%\ _{\pm 3.16}$ | $+173.20\%\ _{\pm 9.80}$ | $+742.47\%\ _{\pm 54.30}$ |
| Entropy | $+3.51\%\ _{\pm 15.95}$ | $+0.04\%\ _{\pm 3.14}$ | $+2.85\%\ _{\pm 3.28}$ | $+32.93\%\ _{\pm 2.67}$ |
| $\text{Entropy}_D$ | $+89.20\%\ _{\pm 14.58}$ | $+0.08\%\ _{\pm 4.24}$ | $+2.98\%\ _{\pm 2.34}$ | $+37.65\%\ _{\pm 2.81}$ |
| LeastConf | $+0.77\%\ _{\pm 14.70}$ | $+2.01\%\ _{\pm 3.99}$ | $+0.06\%\ _{\pm 0.56}$ | $+0.68\%\ _{\pm 0.07}$ |
| $\text{LeastConf}_D$ | $+83.16\%\ _{\pm 14.59}$ | $+0.31\%\ _{\pm 4.38}$ | $+2.73\%\ _{\pm 3.08}$ | $+36.59\%\ _{\pm 2.72}$ |
| Margin | $+0.66\%\ _{\pm 14.53}$ | $+1.85\%\ _{\pm 2.56}$ | $+1.26\%\ _{\pm 2.39}$ | $+0.65\%\ _{\pm 0.19}$ |
| $\text{Margin}_D$ | $+84.00\%\ _{\pm 13.94}$ | $+1.69\%\ _{\pm 3.90}$ | $+3.95\%\ _{\pm 3.93}$ | $+38.88\%\ _{\pm 2.49}$ |
| SupLoss$^\star$ | $+3.04\%\ _{\pm 14.99}$ | $+1.54\%\ _{\pm 6.26}$ | $+1.61\%\ _{\pm 1.84}$ | $+1.36\%\ _{\pm 1.34}$ |
| KAL | $+2.52\%\ _{\pm 13.69}$ | $+0.32\%\ _{\pm 2.56}$ | $+6.43\%\ _{\pm 2.02}$ | $+23.42\%\ _{\pm 0.21}$ |
| $\text{KAL}_D$ | $+88.28\%\ _{\pm 20.79}$ | $+5.41\%\ _{\pm 6.23}$ | $+8.18\%\ _{\pm 1.29}$ | $+62.98\%\ _{\pm 2.37}$ |

# B  SOFTWARE

```python
1   # Knowledge-drive Active Learning - Experiment on the XOR problem
2   tot_points = 10000
3   first_points = 10
4   n_points = 5
5   n_iterations = 198
6   seeds = range(5)
7   x = np.random.uniform(size=(tot_points, 2))
8   y = ((x[:, 0] > 0.5) & (x[:, 1] < 0.5)) |
9       ((x[:, 1] > 0.5) & (x[:, 0] < 0.5))
10  x_train, x_test, y_train, y_test = train_test_split(x, y, test_size=0.1)
11
12
13  # Defining constraints as product t-norm of the FOL rule expressing the XOR
14  def calculate_constraint_loss(x_continue, f):
15      # discrete_x = (x_continue > 0.5).float()
16      discrete_x = steep_sigmoid(x_continue).float()
17      x1 = discrete_x[:, 0]
18      x2 = discrete_x[:, 1]
19      cons_loss1 = f * ((1 - (x1 * (1 - x2))) * (1 - (x2 * (1 - x1))))
20      cons_loss2 = (1 - f) * (1 - (1 - (x1 * (1 - x2)) * (1 - (x2 * (1 - x1)))))
21      return cons_loss1 + cons_loss2
22
23  # Constrained Active learning strategy
24  # We take the p elements that most violates the constraints and are among available
            idx
25  def cal_selection(labelled_idx, c_loss, n_p):
26      c_loss[torch.as_tensor(labelled_idx)] = -1
27      cal_idx = torch.argsort(c_loss, descending=True).tolist()[:n_p]
28      return cal_idx
29
30  net = MLP(2, 100)
31  accuracies = []
32  used_idx = randint(0, x_train.shape[0], first_points).tolist()
33  available_idx = [*range(tot_points)]
34  for n in range(n_iterations):
35      train_loop(net, x_train, y_train, used_idx)
36
37      with torch.no_grad():
38          preds_train = net(x_train).squeeze()
39          preds_test = net(x_test).squeeze()
40      accuracy = accuracy_score(preds_test, y_test)
41      cons_loss = calculate_constraint_loss(x_train, preds_train)
42
43      available_idx = list(set(available_idx) - set(used_idx))
44      active_idx = cal_selection(used_idx, cons_loss, n_points)
45      used_idx += active_idx
46
```

Listing 1: KAL code - Example on the XOR problem. For simplicity we do not consider in this case a train and a test set.

The Python code and the scripts used for the experiments, including full documentation, is freely available under Apache 2.0 Public Licence in a GitHub repository, and it is also provided in the supplementary material. The proposed approach only requires a few lines of code to train a model following the KAL strategy, as we sketch in the code example reported in Listing 1.

