# OpenReview forum: "Knowledge-Driven Active Learning"
_ICLR.cc/2023/Conference — Submitted to ICLR 2023_

### Official Review · Reviewer_fw63 · 2022-10-22

**Confidence:** 3
**Correctness:** 2
**Technical Novelty And Significance:** 3
**Empirical Novelty And Significance:** 3
**Recommendation:** 5

**Clarity, Quality, Novelty And Reproducibility:**

Clarity: Not coming from a first-order logic background, I needed to read extra literature to understand the paper. This is totally fine for reviewers but if the paper is oriented towards a general AL audience, putting more background information of how to convert FOL through T-norm and the limitations of such conversion is crucial.

Also, the computation time advantage is heavily highlighted; I would suggest making room for a more detailed explanation about how and what specific knowledge is injected into each of the datasets and how new researchers can come up with similar knowledge if they would like to use KAL. The main reason I think time complexity is not an issue for Active Learning is because of AL's innate assumption of expensive labeling (time/money wise) and therefore, ten-fold increase in the sampling process might still not be the bottleneck.

Quality: The overall quality of the paper is good

Novelty: This paper is novel to the best of my knowledge

Reproducibility: Code is provided with good documentation.

**Details Of Ethics Concerns:**

This work potentially brings (or, equally, mitigates) significant bias towards the machine learning system due to the subjective knowledge injection into the model with First order logic. If such a system is used in an unfair way, e.g., biased judgment injected, in high-stake systems, it might pose potential ethical concerns to society. Also, the combination with XAI further complicates the "bias" as XAI is usually used in higher stakes systems due to its explainability (often trading off pure performance compared to black-box models) and humans might not notice what the influence is injected through the "knowledge" injected by XAI to KAL.

**Strength And Weaknesses:**

Strength:

This paper proposed an interesting idea: using domain knowledge to improve active learning performance. The benchmarked traditional AL methods are plentiful, and experimental results show the promise of KAL. The ablation study shows the amount of knowledge added to the AL progress. The graphic looks and clean.

Weaknesses:
Apart from some clarity issue (below section), I would like to question the experimental design of this work, especially on the XOR illustration that serves the main purpose to conveying how this method work to the audience:

The initial batch of samples of the AL process is not random, or at least not representative of random. If we look at Figure 1, the first iteration where the blue points are previous samples (and therefore the initial samples that might be the same across experiments), they are heavily biased towards the top left corner. This is an important issue as the majority of the traditional AL algorithm relies on a randomly sampled initial set like BALD. If we then look at Figure 3, the BALD and Margin result, non of the queried points are in the bottom right corner, which indicates the model has not even seen any of those points. The only explanation that made sense to me is that their initialization is biased towards the top left corner, just like Fig 1 shows. This raises strong concerns with respect to the experimental setup.


**Summary Of The Paper:**

The paper introduces the idea of injecting human knowledge into the active learning process by First order logic using Triangle Norms. By formulating domain expert knowledge as loss functions (acquisition function), unlabeled points can be selected based on the number of violated knowledge. The paper tested on 4 different datasets and compared with a bunch of existing AL classification tasks and showed that KAL is surpassing average traditional AL methods.

**Summary Of The Review:**

Given the concerns I have around the experimental setup, along with the presentation issue of the storyline for AL audiences, I recommend borderline reject for this manuscript.

---

> ### Author Response · Authors · 2022-11-18
> **Answer Reviewer fw63**
>
> We thank you a lot for having appreciated our work and in particular the novelty, the presentation, and the originality of the proposed method.
> In the following, we try to address the issues that you have highlighted.
> 1. Thank you very much for raising the point that the initial batch of samples in the XOR experiment does not seem to be random. It actually is, it is one of the 10 random samplings from which we started (identical starting set for all methods) to actively label samples. This phenomenon happens a few times, which explains the poor results of uncertainty methods in this task. When the initial random sampling is not covering completely the 4 data distributions, KAL can discover the distributions not covered by the initial sampling, unlike uncertainty-based methods.  However, we agree that it was not sufficiently stressed. For this reason, in the revised version of the paper you can see we specified clearly that the initialization comes from a random sampling in different places (e.g., in the caption of Fig.1). Also, we rewrote entirely the two paragraphs commenting on this experiment (Sec. 2.2 and Sec 3.3). We think that now it should be sufficiently clear that the initial sampling is random (and identical for all methods), and that sometimes the initial random sampling does not cover all data distributions.
> 2. Thank you also for noticing that incorporating some background on FOL and T-Norms could help readers fully understand the paper. We have added more details and examples in the main paper (See Section 2.1) and an entire appendix of background on FOL rules and on their conversion by means of T-Norms (Appendix A.1). We think that the paper is now easier to comprehend also for people without much background on logic. Also, we added in the conclusion a possible limitation of FOL.
> 3. We agree with the reviewer that computational time may not be of crucial importance in active learning. Indeed, the focus is on the cost of labelling. However, if we consider active learning as a process where an expert is required to iteratively label batches of data and retrain a model, all the steps in the pipeline should be as efficient and time-saving as possible to also save expert time (interactive Machine Learning, iML[1], is getting more and more attention even if the seminal works dated from 2000-2001). Therefore, we still think that being computationally non-expensive is a nice feature to have.
> 4. We are very much glad that you pointed out that this work may be involved with bias issues. Indeed, this is exactly why we think that domain experts may be more willing to use our method w.r.t others: the biases either already present in the data or that they may put on the model are explicit with KAL. This is also true when using KAL in conjunction with an XAI method: what the network is learning is explained by the XAI method to the expert who explicitly sees why labelling a certain sample (i.e. which rules the network violates on that sample). On the contrary, the biases induced in the model by the samples chosen with standard AL techniques, are non-visible and this is one of the reasons why Deep Learning is seen as a black box. In future work, we are already working on whether it is possible to de-bias a model by selecting those samples that violate the biases that the model has learnt. Thank you again from getting this aspect of our work.
>
> [1] Wondimu, N. A., Buche, C., & Visser, U. (2022). Interactive Machine Learning: A State of the Art Review. arXiv preprint arXiv:2207.06196.
>
> We hope to have correctly coped with all issues that you correctly raised. In case we have not, we hope you will give us the chance to better clarify the remaining doubts. Otherwise, we hope you will  raise your mark to allow the full acceptance of the paper: it would mean a lot to us to present this paper in such a prestigious venue.

---

### Official Review · Reviewer_JCBB · 2022-10-23

**Confidence:** 5
**Correctness:** 4
**Technical Novelty And Significance:** 3
**Empirical Novelty And Significance:** 4
**Recommendation:** 8

**Clarity, Quality, Novelty And Reproducibility:**

**Quality**:
The paper is in general of good quality, however, as stated above, I found some of the statements imprecise/misleading. More in detail:
1. Expressivity of the constraints: since the constraints considered in the framework are transformed using the t-norms the framework proposed in the paper cannot handle full FOL (with functions and infinite domains). Since all the constraints studied involve relationships between data and classes the authors should talk about propositional logic constraints.
2. The authors should state since the introduction that they are confining their method to single-label or multi-label classification problems (they cannot handle regression).
3. The authors write that they can have constraints over the input data, however this makes the assumption that the features are binary (as they are then transformed with the sigmoid function). This is then reflected in the XOR-like problem, where the authors write the constraints where the atoms are algebraic functions. This can be misleading as in general their framework does not support such expressivity. (Please correct me if I am wrong, suppose I have the feature "Age" can your framework incorporate the constraint "if Age > 100 then..."? )
4. The authors talk about the Monte-Carlo Dropout version  of their model, without ever explaining exactly what it does. There is a reference to another paper, but each work should be intra-contained, so it would be useful to have some lines explaining the differences between KAL and $\text{KAL}_D$.
5. The authors write: "By always selecting the data that violate this corpus of rules, KAL ensures them that the trained model respects the provided knowledge also at test time." I don't see how this can be true. In general it is not enough to just "provide a datapoint" to a model to guarantee the satisfaction of the constraints. In this respect, in the recent years there have been a some  models that incorporate the constraints into the loss/topology of neural models to guarantee their satisfaction (see e.g., [1,2,3]). It would be nice to have a discussion on how they could be used together with KAL to actually get the satisfaction of the constraints by design.
6. Why is the ablation study reported in table 2 done only on CUB200? Could you do it for all datasets?
7. Table 3: Instead of reporting the violations computed as the increased percentage over the violation of a model trained to respect this knowledge, could the number of violations be directly reported? It would be more informative and less misleading. *As a side node:* why is KAL here called $\text{KAL}$ with *small* as subscript? What is the difference between the this and normal KAL? Why is the set of constraints called $\mathcal{K}$ with *CUB-S* as subscript$? What does the "-S" stand for?
8. In Figure 5 KAL is the only one that is provided with orange datapoints around the point (1.0,0.0) at iteration 0 why is that the case? Were those datapoints provided by the authors? Shouldn't all the models start with the same training set? Why is it not the case? Can the authors ensure that all the models were provided with the same set of labelled datapoints at iteration 0?

Another major point that undermines the findings done by the authors is that they considered one way to include the logical constraints in the training process (i.e., transforming the constraints using the t-norms). However, there has been quite some development in the recent years in the field of "deep learning of logical constraints" (see survey [4] ) and it would have been useful to at least discuss why the t-norm method has been chosen over others. However, most of the recent methods are not even mentioned. For example, in [6] the Semantic Loss has been proposed. The semantic loss has the advantage (over the t-norms) of being syntax dependent. Why are the t-norms convenient wrt such method? Are you planning to try this out in the future?

Another concern that I have is what happens when we have partial knowledge? Suppose that in the XOR example we have only the domain knowledge: $$
y(x) = 1 \text{ if } x_1 > 0.5 \wedge x_2 \le 0.5
$$
my guess is that KAL would only learn to output one only for the datapoints such that $ x_1 > 0.5 \wedge x_2 \le 0.5$. Would you mind actually providing the results for this case? My guess is that for each iteration we would need to also select some datapoints at random. However, it would be interesting to study what the model learns when we vary the percentage of such randomly chosen datapoints.
Since it is not a resource heavy experiment, would you mind running it and reporting the results?

**Clarity:**

The paper is very clear and easy to follow.

**Novelty:**
The paper is very novel and potentially very relevant

**References:**

[1] Paolo Dragone, Stefano Teso, and Andrea Passerini. Neuro-symbolic constraint program- ming for structured prediction. In Proc. of IJCLR-NeSy, 2021.

[2] Eleonora Giunchiglia and Thomas Lukasiewicz. Multi-label classification neural networks with hard logical constraints. JAIR, 72, 2021.

[3] Nicholas Hoernle, Rafael-Michael Karampatsis, Vaishak Belle, and Kobi Gal. MultiplexNet: Towards fully satisfied logical constraints in neural networks. In Proc. of AAAI, 2022.

[4] Eleonora Giunchigla, Mihaela C. Stoian, and Thomas Lukasiewicz. Deep learning with logical constraints. In Proc. of IJCAI, 2022.

[5] Jingyi Xu, Zilu Zhang,Tal Friedman, Yitao Liang, and Guy Van den Broeck. A semantic loss function for deep learning with symbolic knowledge. In Proc. of ICML, 2018.


**Strength And Weaknesses:**

**Strengths:**
1. The proposed approached is very novel. As far as I know, nobody has proposed the usage of logical constraints for the task of active learning.
2. The authors propose synthetic experiments that help in understanding the strengths of the work.
3. The problem has been analysed from a variety of different angles.

**Weaknesses**:
Even though I really liked the paper, I found that at times it was written in an imprecise/misleading way. Please see the box below for more details.

**Summary Of The Paper:**

The paper proposed a new methodology for active learning. In particular, they propose to exploit the background knowledge available about a problem to better select the datapoints to be labelled. The background knowledge must be expressed as logical constraint as it is then mapped into the loss function via the usage of the t-norms. An extensive experimental analysis has been done to evaluate the strengths and weaknesses of the model.

**Summary Of The Review:**

TL;DR I think this paper has all the potential to be a great paper. However, in the current form I have too many concerns. If all of them are addressed I would happily fully accept the paper

---

> ### Author Response · Authors · 2022-11-18
> **Answer to Reviewer JCBB (2)**
>
> 7. Regarding Table 3 we have not reported the number of violated rules since these are  soft constraints not hard ones, so the violation of each rule is a fuzzy value between 0 and 1. The sum (or the average) of the constraint violation over the provided knowledge and over the test data is therefore not a meaningful value per se, but only when compared to the violation of other strategies. We do not exclude, however, that there might exist other ways to report this result. Regarding the name KAL_{SMALL}, it is named as such because it is not using the whole knowledge on CUB but only a portion of it and therefore the name $\mathcal{K}_{CUB-S}$ where S stands for small. We better write it in the paper as well.
> 8. Regarding the initialization of the experiment on the XOR, all methods are provided with the same identical initialization. Starting points are always randomly selected and we reported one of the 10 random initialization of the experiment (illustrated by the blue points in the leftmost figures of Figure 1-3-5). KAL as the other strategies is not provided with points around (1.0, 0.0) at iteration 0, but it is capable of selecting them at the first iteration since they are the points violating more the provided knowledge (orange points). For this reason we argue that KAL is capable of discovering novel data distribution unlike uncertainty methods.  However, we agree that it was not sufficiently clear in the paper. For this reason, in the revised version of the paper we specified clearly that the initialization comes from a random sampling in different places identical for all methods (e.g., in the caption of Fig.1). Also, we rewrote entirely the two paragraphs commenting on this experiment (Sec. 2.2 and Sec 3.3).
>
> Regarding the related works on learning with constraints, we thank the reviewer for pointing out this method.  We are indeed well aware of the interesting Semantic Loss work and of different ways of linking symbolic constraints with learning models. We thank you for suggesting the survey: we have added it as a reference for a complete and updated list of works in the learning with constraints field. Indeed, there exist other neurosymbolic methods that could be applied to the active learning domain. However, we chose to employ the SBR framework [1] (based on t-norms) because it is first of all very flexible: given any kind of rule (even very complex ones), the translation into a loss function is extremely straightforward and can be even automatized as shown in [2]. On the contrary, the Semantic Loss method requires building specific logic circuits corresponding to the rules by means of circuit compilation techniques. Also, it is more general: it can handle full FOL as shown previously, differently from other approaches and, namely, Semantic Loss which cannot handle existential quantifiers as also reported by the survey itself. Finally, it really seems that they are semantic-independent because they actually fixed the structure of the logic rule: they always use CNF, which, even when using T-norm it results to be semantic-independent.
>
> Regarding the partial knowledge on the XOR problem, we confirm that when dealing with partial knowledge, the performance deteriorates. More precisely, we experimented that when using $x_1 \land \neg x_2 \leftrightarrow f$ instead of the complete rule, the performance of an active learning strategy equipped with such knowledge decreases to an AUBC of 87.88, slightly lower than the average of uncertainty-based techniques. We would like to highlight, however, that this rule does not hold true for a large portion of data. Regarding the possibility of randomly selecting some samples to couple with partial knowledge, we can confirm that it would be an effective solution. We already adopted it when coupling the proposed method with XAI techniques as reported in Section 3.5. In this case we employed 40 % of randomly selected samples which allowed the XAI method to gradually expand the extracted knowledge, indeed, from $x_1 \land \neg x_2 \leftrightarrow f$ to $x_1 \oplus x_2 \leftrightarrow f$.
>
> We hope to have correctly coped with all issues that you correctly raised. In case we have not, we hope you will give us the chance to better clarify the remaining doubts. Otherwise, we hope you will raise your mark to allow the full acceptance of the paper: it would mean a lot to us to present this paper in such a prestigious venue.
>
>
> [1] M. Diligenti, M. Gori, and C. Sacca. Semantic-based regularization for learning and inference. Artificial Intelligence, 244:143–165, 2017.
>
> [2] G. Marra, F. Giannini, M. Diligenti, and M. Gori. Lyrics: A general interface layer to integrate logic inference and deep learning. In ECML/PKDD, 2019.
>
> [3]  Corbiere, C., Thome, N., Saporta, A., Vu, T. H., Cord, M., & Perez, P. (2021). Confidence estimation via auxiliary models. IEEE Transactions on Pattern Analysis and Machine Intelligence.

---

> > ### Comment · Reviewer_JCBB · 2022-11-30
> > **Thanks for the detailed answers**
> >
> > Dear authors,
> >
> > thanks for the detailed answered. I was very happy to hear that the review helped in improving the paper (and indeed it reads much better!).
> >
> > Just as a further clarification: in appendix A (or even better in the main body of the paper) please state that since you have only used universally quantified constraints, then these can be also expressed in propositional logic. This is important because it will allow future researchers to pick the expressivity of the constraints they want to work with, and I believe that most of the constraints actually used nowadays in ML can be expressed in propositional logic, as the existential quantifier is very rarely used. (More expressivity is not always a feature, as it comes with higher complexity)
> >
> > I trust that you will make this minor modification, and I have raised the score to a full accept.
> > Congratulations!

---

> ### Author Response · Authors · 2022-11-18
> **Answer to Reviewer JCBB**
>
> We thank you very much for having appreciated the paper and its novelty. Also, we would like to thank you for your extremely thorough review. We think we have learnt and improved the paper a lot by following your suggestion. In the following, we try to answer the issues that you have highlighted:
>
> 1. We thank you for noticing that it may have seemed that our framework did not support FOL. Indeed, we only used simple relationships between input data and classes that could have been dealt with only Propositional Fuzzy Logic. However, our framework does fully support the potentiality of First-order fuzzy logic (with existential quantifiers and logic functions among different domains) in those contexts where it is required. We have added further details and examples in the main paper and a new appendix only dedicated to FOL rules and how to convert them by means of T-Norms (Appendix A.1). To better stress this point, we also have improved and extended our reference in Sec. 2.1 and A.1 to [Marra2019], where this is fully and better explained. We think that our claim is now better and more clearly motivated.
> 2. We thank you for the criticism on regression tasks. At the beginning of Section 3 we had written that we were focusing on classification tasks only in this work. However, we did not exclude the possibility of applying the proposed method on regression task, because we knew that it was possible (even though we did not experiment it). Your comment, however, has motivated us in performing an experiment, showing that KAL can be effectively employed in regression tasks, unlike uncertainty methods unless an auxiliary model dedicated to predict the confidence of the regression model is involved [3]. We have added a new appendix (Appendix A.2) showing that KAL results again in the best strategy among the compared ones. In this scenario, you only need to define the logic predicates over intervals of the output (e.g., charges(x) > 10000), which can be easily done by using a sigmoid centered at the desired value.
> 3. We thank you for noticing that it may have seemed that, as stated previously, our framework could only deal with binary inputs. As you may have probably noticed from the previous point, our framework can handle algebraic functions over the input or the output. To do so, you only need to center the sigmoid at the desired value, e.g. to assess the validity of $\mathbf{age} > 100$ the sigmoid should be $\frac{1}{1 + e^{-(x - 100)}}$. In the regression task, indeed, we used (among others) the rule $\neg \mathbf{smoker} \land \mathbf{age} > 40 \leftrightarrow \mathbf{charges} > 7500 \land \mathbf{charges} < 15000 $.
> 4. We thank you for noticing that we did not mention MonteCarlo Dropout works in active learning. We have added a sentence in the compared method to explicitly say how it works.
> 5. We thank you very much for noticing this mistake. Indeed, we do not ensure domain experts that the domain knowledge is respected in general, but only more than other active learning methods. It was a badly written sentence, the true statement regarding this paragraph was reported at the end of the paragraph: “it ensures domain experts that the provided knowledge is respected significantly more than using […] standard active learning strategies”. We reformulated the sentence. Regarding the cited literature, we thank you very much for pointing them out. The core principle of this active learning method is however on having soft-constraints that are violated by the network and this drives the labelling process. If a model is trained to fully satisfy a given knowledge (e.g. through hard constraints), it seems to us it might be difficult to employ the proposed active learning scenario since they would provide zero constraint violation on labelled data.
> 6. Regarding the ablation study on the quantity of knowledge, we provided it only on the CUB200 dataset, since it is the only task on which the amount of knowledge is sufficiently large (313 rules). In this case, indeed, selecting only a portion of rules still allows to retain a sufficiently large corpus of rules to be meaningful. However, for the sake of completeness, we are adding the complete ablation study in Appendix A7. The complete table confirms the result reported for the CUB200: in all datasets, the amount of knowledge is proportional to performance improvement.

---

### Official Review · Reviewer_KFBz · 2022-10-25

**Confidence:** 2
**Clarity, Quality, Novelty And Reproducibility:** I believe that clarity and novelty ar…
**Correctness:** 3
**Technical Novelty And Significance:** 2
**Empirical Novelty And Significance:** 2
**Recommendation:** 6

**Strength And Weaknesses:**

**Strengths**

* This paper presents an interesting and flexible framework for active learning. The proposed approach allows users to specify domain information in the form of FOL to determine selection criteria.
* The authors demonstrate empirical effectiveness via a series of empirical experiments . The experiments are extensive and used to validate the approach.

**Weaknesses**

* I think that the paper could be improved if there was more succinct understanding as to where and when the proposed approach is more effective or efficient than baseline methods in active learning
* Table 1 leads to a somewhat confusing landscape of empirical success, where baseline methods of kcenter and kmeans are highly effective. I think the presentation can be improved by highlighting reasons to select a particular method more.

**Summary Of The Paper:**

This paper presents an approach for active learning that selects examples using logic constraints specified as domain knowledge. The proposed approach is evaluated in a series of empirical experiments compared to traditional active learning baselines.

**Summary Of The Review:**

An interesting approach for using domain knowledge in active learning. The proposed approach is extensively evaluated against active learning methods, however, in places where the best method is not abundantly clear, the paper could be improved to specify why one approach is preferred to another more succinctly.

---

> ### Author Response · Authors · 2022-11-18
> **Answer to Reviewer KFBz**
>
> We thank you very much for having appreciated the effectiveness, flexibility, and novelty of our proposal. In the following, we try to address the issues that you have raised:
> 1. Thank you very much for suggesting that it was not clear in which scenario KAL is more effective. We have added a paragraph at the end of the Sec. 3.1 and a few sentences in the conclusion, precisely with this scope. We think that the take-home message regarding where and when to use KAL should be clearer. As a more general answer to that point, KAL can always be considered more effective than baseline methods whenever sufficient domain knowledge is available since no other active learning technique benefits from external knowledge.
> 2. We agree that the effectiveness of some baselines (particularly of KCENTER) is surprising. However, it only happened in the simple XOR and IRIS datasets: on the Computer vision tasks (due to the complexity of the data space to model) KCENTER dramatically reduced the network performance (up to -15% on the Animals dataset w.r.t KAL). We have added a comment in the experimental section to clarify where and why KCENTER works well. KMEANS instead does not seem to be an effective technique, being overcome in three datasets out of four by random sampling.

---

### Official Review · Reviewer_r58Q · 2022-10-25

**Confidence:** 3
**Correctness:** 3
**Technical Novelty And Significance:** 2
**Empirical Novelty And Significance:** 3
**Recommendation:** 6

**Clarity, Quality, Novelty And Reproducibility:**

Paper is clear and source code investigation does not raise many questions regarding reprodubibility.

**Strength And Weaknesses:**

## Strengths:

Authors propose a light-weight model for active learning incorporating domain knowledge using the T-Norms based rules introduced in [1].


Results of the proposed model are convincing as a result of the multiple tasks and variegated baselines employed for state-of-the-art comparison.


Paper is well written and well organized.



## Weaknesses:

The proposed method is applied to classification problems and in somewhat balanced data contexts. It would be interesting to check the power of domain knowledge in data imbalanced contexts.

 Proposed model is somewhat novel as the crux of the FOL encoding is based on the previously proposed T-Norm idea.

The discussion about contexts in which the proposed method would out-perform baselines would be especially insightful to readers.

Incorporating a little more background regarding FOL might help readers fully comprehend the paper better.

## References:

1. Klement EP, Mesiar R, Pap E. Triangular norms. Springer Science & Business Media; 2013 Apr 17.

**Summary Of The Paper:**

Summary: Authors propose an active learning pipeline that incorporates domain knowledge into the pipeline for more efficient active learning & yielding models more consistent with known domain rules. Specifically, the model employs first-order-logic (FOL) based encoding to employ domain rules to inform examples to be selected for active learning. Authors demonstrate the effectiveness of their generic FOL pipeline on variegated tasks like Object recognition, classification and compare their model to various state-of-the-art uncertainty based active learning approaches and other clustering based methods. Overall the proposed method yields good results.





**Summary Of The Review:**

The proposed model is somewhat novel and the core idea (i.e., encoding domain knowledge as FOL using T-Norms) is based on previously published paper.

---

> ### Author Response · Authors · 2022-11-18
> **Answer to Reviewer r58Q**
>
> We thank you very much for having appreciated the effectiveness, the presentation, and the low computational cost of our proposal. In the following, we try to address the issues that you have raised:
>
> 1. Thank you for suggesting applying our method to imbalanced contexts: we will surely experiment with this scenario in future work. We are confident that our method will work nicely since it selects data violating the given knowledge regardless of data distribution. Therefore, as soon as the knowledge concerning less-represented data is violated, more points from that distribution will be chosen for labelling. And this may happen even when this distribution is not covered by the starting random sampling, as shown in Sec. 3.3. Also, the diversity in the violated rules ensures that a diverse batch of data is selected.
> 2. Regarding the modest novelty, we do not fully agree on that point, even if it may have been our fault because we might not have stressed it enough. Indeed, FOL encoding and T-Norm theory have been already used in previous works, but as far as we know it is the first time that symbolic constraints (using FOL) have been involved in an active learning strategy for non-symbolic models. We think this work can open an important research pathway toward bridging the gap between “black-box models” and expert knowledge.
> 3. Thank you very much for suggesting that it was not clear in which scenario KAL is more effective. We have added a paragraph at the end of the Sec. 3.1 and a few sentences in the conclusion, precisely with this scope. As a more general answer to that point, KAL can always be considered more effective than baseline methods whenever sufficient domain knowledge is available since no other active learning technique benefits from external knowledge.
> 4. Thank you also for noticing that incorporating some background on T-Norms and FOL could help readers fully understand the paper: we have added more details and examples in the main paper and a new appendix only dedicated to FOL rules and their conversion by means of T-Norms (Appendix A.1). We think that the paper is now easier to comprehend.

---

### Official Review · Reviewer_NXVX · 2022-11-01

**Confidence:** 4
**Clarity, Quality, Novelty And Reproducibility:** The paper is well-motivated, easy to …
**Correctness:** 3
**Technical Novelty And Significance:** 4
**Empirical Novelty And Significance:** 3
**Recommendation:** 8

**Strength And Weaknesses:**

Active learning is a machine learning paradigm where a subset of examples is selected from a large pool of unlabelled data to query their labels. Most of the active learning literature works are based on either uncertainty or diversity, or both. I find the work novel because it introduces the idea of injecting domain knowledge.

The paper is well-motivated, and the previous works are discussed adequately. However, some of the recent works on model-based active learning are missing:

Sinha, Samarth, Sayna Ebrahimi, and Trevor Darrell. "Variational adversarial active learning." Proceedings of the IEEE/CVF International Conference on Computer Vision. 2019.

Yoo, Donggeun, and In So Kweon. "Learning loss for active learning." Proceedings of the IEEE/CVF conference on computer vision and pattern recognition. 2019.

Caramalau, Razvan, Binod Bhattarai, and Tae-Kyun Kim. "Sequential graph convolutional network for active learning." Proceedings of the IEEE/CVF conference on computer vision and pattern recognition. 2021.

The experiments are performed for both image classification and object recognition. The experimental results are either comparable to better than the existing method without incurring much computational cost.  Although the experiments are done on a small scale, I think this should not be considered a weakness. However, I wondered how domain knowledge could be applied to active learning for regression tasks.


The paper is well presented, and explaining with the toy example makes the idea easy to understand.


**Summary Of The Paper:**

This paper presents a novel idea of incorporating domain knowledge into first-order logic  (FOL) for active learning. Consistency between the predictions made on unlabelled data and the quantified domain knowledge encoded in the form of FOL is assessed, and the examples defying make the candidate examples to query their label. Experiments are performed in multiple data sets and compared with a wide range of existing methods.


**Summary Of The Review:**

I find the idea of injecting domain knowledge into active learning very interesting. Although the experiments are done on a small scale, they are convincing.

---

> ### Author Response · Authors · 2022-11-18
> **Answer to Revewer NXVX**
>
> Thank you very much for having fully appreciated our work. Your review has been really motivational for us.
> Concerning the regression task, thank you for your suggestion: KAL can be effectively employed in this scenario. We have added a new appendix (Appendix A2) showing that KAL can be applied to regression tasks, unlike uncertainty methods (unless using an external method to predict model confidence as in [1]), and it results again the best strategy among the compared ones. In this scenario, you only need to define the logic predicates over intervals of the output (e.g., charges(x) > 10000), which can be easily done by using a sigmoid centered at the desired value.
> Concerning the missing previous works, thanks for pointing them out. We have added Sinha 2019 and Caramlau 2021 in the Related Work (Yoo 2019 was already cited among the compared methods since we compared against a simplified version of this method).
>
> [1]  Corbiere, C., Thome, N., Saporta, A., Vu, T. H., Cord, M., & Perez, P. (2021). Confidence estimation via auxiliary models. IEEE Transactions on Pattern Analysis and Machine Intelligence.

---

### Author Response · Authors · 2023-02-08
**Not understandable Final Decision**

The final decision for this paper is completely not understandable. We have never ever argued with a decision of Area Chairs on paper acceptance: However, this case is extraordinary.
First, this paper has received 5 reviews with an average score of 6.7 with two reviewers assigning a score of 8 with high-confidence. Furthermore, the comment on the final decision is objectively wrong and in contrast with the comments made by all 5 reviewers.

* AC: “Applicability is limited beyond simple dataset […] Empirical evaluation is run against various active learning methods on 3 datasets”.
    -  We performed experimental evaluation on 6 datasets (2 tabular data, 2 computer vision, 1 object recognition, 1 regression). On the CUB dataset (200 classes and over 10K images) the proposed method results the best one.
    -  R1: “Experiments are performed in multiple data sets and compared with a wide range of existing methods”.
    -  R2: “Results of the proposed model are convincing as a result of the multiple tasks and variegated baselines employed”.
    -  R3: “The experiments are extensive and used to validate the approach.”
    -  R4: “The problem has been analysed from a variety of different angles.” […] “An extensive experimental analysis has been done to evaluate the strengths and weaknesses of the model.”
    -  R5: “The benchmarked traditional AL methods are plentiful, and experimental results show the promise of KAL”
* AC: “The reviewers believed writing and clarity can be improved”:
    -  R1: “The paper is well presented, and explaining with the toy example makes the idea easy to understand.[…] The paper is well-motivated, easy to understand, and novel.”
    -  R2: “Paper is well written and well organized.”
    -  R3: “Clarity and novelty are adequate.”
    -  R4: (after rebuttal) “I was very happy to hear that the review helped in improving the paper (and indeed it reads much better!).”
* AC: “Scalability of the proposed method is limited. […] No discussion on the search or its cost is provided.”
    -  We provided an entire subsection (Sec. 3.7 “KAL is not computationally expensive”) where we show that the computational cost of the proposed method is among the lowest between the 14 compared methods. We stress this point even in the abstract: “its computational demand is low, unlike many recent active learning strategies.”.
    - R1: “The experimental results are either comparable to better than the existing method without incurring much computational cost.”
    - R2: “Authors propose a light-weight model for active learning”.
    - R5: “The computation time advantage is heavily highlighted”.
* AC: “Limited novelty”:
    - R1: “I find the work novel because it introduces the idea of injecting domain knowledge”
    - R4: “The proposed approached is very novel. […] The paper is very novel and potentially very relevant”
    - R5: “This paper proposed an interesting idea […] This paper is novel to the best of my knowledge”
* AC: “There is a wide variance among reviewer scores”.
    - 2 reviewer gave us 8 with very high confidence, 2 reviewer 6, 1 reviewer 5.

As a matter of respect, primarily towards the 5 reviewers who spent much time in reviewing the paper and who were convinced on its quality, we think that this decision should be at least double-checked.
We thank you very much for the time that you will take to revise this decision.

---

### Decision · Program_Chairs · 2023-01-20

**Decision:**

Reject

**Justification For Why Not Higher Score:**

- Scalability of the proposed method is currently limited: It appears to be a brute-force search with model evaluation across the entire unlabeled set. While this is fine for the small scale datasets considered in the paper, it would not scale to large unlabeled data commonly encountered in real applications. Having a larger model further exacerbates the cost. Applicability is therefore limited beyond simple datasets.
- Improvements of the proposed method are often marginal (as pointed out by reviewers) or not significant (e.g. Table 1). Furthermore, whereas the random baseline doesn't suffer from O(n) runtime costs, gains of the proposed method over it are only marginal. In their comments, authors also indicate that gains could be marginal compared to random and more investigation is needed to understand when the proposed approach will be beneficial.
- Correctness of evaluation with regard to baselines is unclear. For instance, from the code provided, it seems that the timings of the random baseline are wrongly evaluated using an unnecessary O(n) runtime cost instead of the correct O(1). Correctness of results such as in Table 5 and 19 are therefore unclear.
- Limited novelty and methodological development: The paper is combining T-Norm based logic encoding with active learning in a straightforward way. While this particular combination is novel, using soft constraints for active learning and the encoding of logic via t-norms are well-known. Other methodological developments, e.g., making the method scalable and efficient compared to baselines are currently missing.
- While the presentation has been improved in the revised manuscript, important aspects are still unclear, e.g., discussion of runtime costs, optimization of eq 1, how numerical conversion is done, etc. Also, how to apply the method to more complex datasets like adding logical constraints at image level. Some statements such as “ensures domain experts that the provided knowledge is respected by the model on test data” are also not supported by the paper and are not yet adapted after rebuttal.

**Justification For Why Not Lower Score:**

N/A

**Metareview: Summary, Strengths And Weaknesses:**

The paper attempts to improve active learning by leveraging domain knowledge. In particular, the paper aims to minimize the amount of labeled data required to train a deep network. In this regard, the authors leverage domain knowledge in form of first order logic constraints and propose to search for examples that violate these constraints the most. Empirical evaluation is run against different active learning methods on various datasets. We thank the authors and reviewers for engaging in discussion towards improving the paper which alleviated various initial concerns of reviewers. However, even after rebuttal, there persists still a wide variance among reviewer scores.

While, on average, reviewers are leaning somewhat towards acceptance, the AC recommends to reject and revise at this point. While the paper contains very promising ideas and directions, it has, in its current form, also shortcomings in terms of clarity and technical contributions. In particular, the limited scalability of the current method, a relatively straightforward application of T-norm based logic encoding, and marginal experimental gains limit the insights that can be gained for active learning. In addition, while clarity improved after rebuttal, the presentation of the method and the experiments is still lacking. For instance, from reading the paper it is not clear how the search or maximization problem in Eq(1) is solved and no discussion on the search or its cost is provided. From the code provided, it appears to be a brute-force O(n) search which includes model evaluations across the entire unlabeled set. Related to the supplied code, it is also currently unclear if baselines are correctly evaluated. Given the points above as well as the more detailed aspects below, the AC does not recommend acceptance at this point. However, the AC encourages the authors to revise and resubmit their manuscript as it contains very promising ideas and directions.

**Summary Of Ac-Reviewer Meeting:**

N/A